# Green synthesized silver nanoparticles from eucalyptus leaves can enhance shelf life of banana without penetrating in pulp

Durr-e- Nayab, Shamim Akhtar *

Department of Botany, University of Gujrat, Gujrat, Pakistan

* shamim.akhtar@uog.edu.pk

## Abstract

Bananas are exposed to serious post-harvest problems resulting in agricultural and economic losses across the world. The severity of problem is linked with the process of rapid ripening and pathogens attack. Such problems have led to economic losses as well as a lower yield of nutritionally rich bananas. The global demand to increase the life span of bananas and their protection from pathogens-borne diseases urged the use of antimicrobial edible coatings of nanoparticles. The present experiment has explored the innovative development of green synthesized nanoparticles from Eucalyptus leaf extract (ELE) to increase the shelf life of bananas up to 32 days from the day of collection. Statistically significant results were recorded (P = 0.05) by applying five different concentrations of silver nanoparticles (AgNPs) in ranges of 0.01–0.05%. Various morphological and physiological parameters such as color, decay, firmness, weight loss, pulp to peel ratio, pH, titrable acidity (TA), phenolic contents, protein estimation, ethylene production, starch content and total soluble sugars were measured in Cavendish banana (Basrai). Bananas treated with 0.01% AgNPs showed maximum control on its ripeness over morphological and physiological changes. The increase in shelf life was in order 0.01%>0.02%>0.03%>0.04%>0.05%> control. Further, AgNPs reduced the process of ripening by controlling ethylene production. The result has also proved the safety of banana consumption by simple removal of banana peel as penetration of AgNPs from the peel to the pulp was not detected. It is recommended to use 0.01% AgNPs to enhance the shelf life of banana without effecting its nutritive value.

## Introduction

Banana (*Musa spp*.) is an important commercial fruit that has a great economic value due to its high consumption demand. It is a balanced diet source rich in various minerals, vitamins and carbohydrates with a little amount of protein, making it favorite global food [1]. Despite its huge demand, banana suffers from post-harvest losses, which is major concern of researchers [2]. Bananas are transported from their production areas to distant locations for marketing and consumption. If post-harvest bananas are not treated appropriately, they become susceptible to damage and degradation during transportation, marketing and storage. Bananas are

**Data Availability Statement:** All relevant data are within the paper and its figure files.

**Funding:** The facilities for the experiment were provided by University of Gujrat, there was no additional external funding received for this study.

**Competing interests:** Authors have no competing interests.

physiologically sensitive to decay after harvest due to continuous change in metabolic processes such as transpiration or respiration rate [3]. Physical injuries and enzymatic actions by microorganism attack, or a combination of these factors can cause damage and degradation. Banana injuries and damage may result in moisture loss due to more surface evaporation [4] as well as microorganisms (fungi, bacteria) attack on injured fruits, causing substantially faster respiration rate than that of healthy bananas. Fruits with faster respiration and metabolic activity result in early storage decay or rots [5]. However, it suffers from many post-harvest nutritional losses which cause physiological and morphological changes such as color change, decay, weight loss, loss of starch, protein and phenolic contents [6,7]. For this reason, number of technologies are in practice to extend the shelf life of bananas by subjecting the controlled environmental conditions as low temperature storage, modify the atmospheric condition of storage and packing but these techniques are highly expensive [8].

Nanotechnology is an advanced field of research that has a vast application in medicine, pharmaceuticals, biochemistry, bio-sensing and biomaterial production by the synthesis of nano devices for its physiochemical properties to overcome the limitations of pre-existing technologies [9]. These nano sized particles improved the approach to produce the desirable consequences by the use of reliable, safe, cost effective and easy methodology. These nanomaterials are used almost in every field of life as in cosmetics, paints, environmental remediation, waste water management, electronic devices and the food industry [10]. Today it is a novel approach in food market to prevent post-harvest losses and increase the shelf life of food [11].

Various chemical, mechanical, physical and biological ways are used for synthesis of silver nanoparticles. These mechanisms are laborious, time consuming, costly and have environmental defects as they are the source of toxic byproducts [12]. So, the use of technically reliable, rapid, low cost and ecofriendly techniques are the current challenge for the synthesis of nanoparticles. Biological synthesis of silver nanoparticle by using enzymes and microbes is an environmentally safe technique, but the green chemistry has more advantage on it being easy, quick, safe, economically effective, environmentally non-hazardous and a single-step method for nanoparticles formation [13].

Green synthesis is more valuable platform for large scale nano sized particles synthesis as silver, gold and titanium by using different plant material such as plant extract, vegetables peels, fruits pulp, bark and fruit peels. These plant mediated synthesis of nanoparticles reduce the expensive, laborious and environment unfriendly mechanisms [14]. Green innovative approach of nanoparticles with below 100 nm dimensions has become the great interest of researchers and scientist to develop ecofriendly cost-effective and antimicrobial agent [15].

Silver is a safe, innocuous and antimicrobial agent which can be used to retard the growth of microbes. Control of ethylene production is another property of silver. Due to these applications, the synthesis of AgNPs have grabbed the attention of various researchers. Besides this, AgNPs have extensive role as drug delivery devices, catalysts, anti-microbial and bio-sensing agents. AgNPs produced from banana peel extract proved cost-effective solution for post-harvest management upto eight days by controlling ethylene production, determining starch content, measuring weight loss effect and acting as antimicrobial agent [7]. Similarly, Eucalyptus spp. is also a cheap source for AgNPs formation as edible coatings [16].

The Eucalyptus plant is cultivated in irrigated areas of Punjab and Sindh in Pakistan as well as other tropical countries and it is facing several allegations due to adverse effects on environmental and ecological conditions. Eucalyptus tree is found in large numbers in Pakistan, it is addressed as soil rendering water table and nutrients depletion country due to its high rate of evaporation and deep root system than water recharge as large amount of eucalyptus cultivation is causing a debate in various countries including South East Asia [17,18]. On the other hand, eucalyptus is accused for allelopathic effect by production of a large amount of toxic

allelochemicals [19,20]. Due to its various negative effects, eucalyptus leaves can be an effective source to produce nanoparticles [21]. The great concern is to develop nanoparticles with dual effects being safe, reliable, and viable treatment on microbial activity and ripening process of bananas in an eco-friendly manner [22].

Present study comprehends the green synthesis of nanoparticles from eucalyptus leaves to manage post harvest losses in banana. The aim was extended to optimize different factors as silver nitrate and ELE to synthesize green route AgNPs for its better assembly and characterization by using various analytical techniques as UV-vis spectroscopy, X-ray diffraction (XRD), scanning electron microscopy (SEM), transmission electron microscopy (TEM) and Fourier transform infrared spectroscopy (FTIR) analysis. These techniques are helpful to describe the size, shape and functional features of nanoparticles [23]. Besides this, green synthesized AgNPs can be helpful in enhancing shelf life of banana in lower concentration without effecting its nutrition.

## Materials and methods

### Biosynthesis of AgNPs from eucalyptus leaves

Young *Eucalyptus* leaves from around 40 years old eucalyptus tree were collected from GT Road near Pindi Bypass, Gujranwala produce silver nanoparticles. Eucalyptus leaves were collected randomly, washed with tap water, and distilled water respectively. The small pieces of leaves were dried in shade to obtain the fine powder. Finally, the leaf powder was autoclaved under the pressure of 15Ib/sq inch and temperature of 121˚C for 5 min [21]. Leaf powder of 5 g was boiled in 500 mL of distilled water (10 mg/mL) for 10 minutes and filtered by Whatman filter paper. The extract was stored at 4˚C for further use. For silver nanoparticle synthesis, 2 mM silver nitrate solution was mixed with ELE (10 mg/mL) in different ratio. The change in color from yellow to dark brown represented the formation of AgNPs [21].

### Optimization of AgNPs

Different factors were studied to check their effect on the synthesis of nanoparticles. Briefly, 0.1 mL of the extract was poured in 0.9 mL of silver nitrate solution (1:9 ratio). Further, different concentrations of extract and silver nitrate solution were mixed until the final volume of 9:1 ratio was attained. After that the reaction mixture was heated at 80˚C for 1 hour with continuous stirring and the synthesis of AgNPs was examined by recording UV–visible spectra (λ 300–600 nm) [24].

### Characterization of AgNPs

To analyze the physiochemical properties of synthesized AgNPs, different analytical techniques were used. **For morphological study,** synthesis of AgNPs was monitored by the change in color of the reaction mixture and for quantitative study, the subsequent reaction mixture was centrifuged at 44000 rpm for 15 minutes to attain pellet. Pellet was washed with distilled water for 2 to 3 times to remove the silver ions. After discarding the supernatant remaining pellet was dried in oven at 80˚C to attain AgNPs for characterization [25].

**UV-VIS Spectrometry.** It is a reliable analytical technique which was used to evaluate the synthesis and functional stability. For this process dried nanoparticles were re-suspended in distilled water by recording spectra between 300–700 nm at 0.1 nm resolution through UV-VIS spectrophotometer (UV-1800 SHIMADZU, Shimazdu, Japan) [26].

**Fourier transform infrared spectroscopy (FTIR).** The FTIR technique was used to study the stability and synthesis of nano-scale silver particle by monitoring the 10 mg dry sample of

AgNPs at resolution power of 4 at range of 500–4000 $cm^{-1}$ infrared through Fourier-transform infrared spectrophotometer (NICOLET iS5, Thermo Scientific, USA) [27].

**X-ray diffraction.**   It was a process to measure the purity and crystalline nature of powder form silver nanomaterial by determining the physiochemical property of crystal component. The diffraction pattern obtained from X ray diffractometer (JDX-3532, JEOL Japan) described the crystalline nature size of AgNPs at 2θ range of 0 to 100 by using the Debye–Scherrer's equation [28].

$$D = \left[ 0.\frac{9\lambda}{\beta} cos\, \theta \right] \tag{1}$$

**Scanning electron microscopy (SEM).**   Morphology of AgNPs (Shape and size of powdered Ag nanoparticles) was measured by SEM (JSM-5910, JEOL Japan) [29].

**Transmission electron microscopy (TEM).**   Internal structure (size, shape and organization) of AgNPs was measured by TEM (JEM-2100, JEOL Japan). For this purpose, liquid nanoparticle mixture was subjected on carbon coated copper grid and allowed to dry at room temperature. It is more valuable than SEM due to its high-resolution power and ability to determine analytical features [30].

**Energy dispersive x-ray analysis (EDX).**   This analytical method was used to identify the desired elements. The spectra of peaks showed the true composition of sample by EDX with SEM (JSM-5910, INCA200 Oxford instruments, UK) [31].

## Nanoparticle's coating applications

Cavendish bananas (Basrai) were collected from banana field Tando Jam, Sindh and each set had six bananas per treatment beside an uncoated (T0 = control) set. Banana is widely grown for research purpose by Pakistan Agriculture Research Council, Islamabad, and were available on request without any permission. The banana was dipped in different concentration of AgNps solution (T1 = 0.01%, T2 = 0.02%, T3 = 0.03%, T4 = 0.04%, T5 = 0.05%) for 5 minutes. Then an experimental set of banana was kept at 25˚C. The data was recorded weekly to study the effect of silver nanoparticles.

**Ripening stages of peel color.**   Banana peel color was the most important morphological character to study the ripeness in banana at different stages. All varieties of bananas were studied weekly according to a standard color scale by using visual sense.1-Green, 2-Pale green, 3-Greenish yellow, 4-Yellow and less green, 5-Full Yellow, 6-Almost yellow, 7- Yellow with light brown flecked, 8-Yellow with full brown areas/Decay [32].

**Decay rate of banana.**   The decay rate of each banana variety was determined in percentage by using a standard decay scale after the interval of each week [32].

**Firmness (N).**   Banana samples firmness was measured by using manual penetrometer (GY-3). Banana finger was placed on plain surface and pushed the penetrometer at its middle portion to take the perfect reading [8].

**Weight loss (%) measurement.**   The difference between initial and final weight was recorded in control and coated set of bananas to check the effect of AgNPs on total weight [22].

**Pulp to peel ratio (%).**   In order to determine the effect of AgNPs on pulp to peel ratio, the weight of pulp was divided by peel weight [22].

**Moistening contents (%).**   Moisture content of banana pulp from un-ripe to ripened stages were determined by placing 3-4mm thick slice in drying oven for 24 hours at 105˚C.

The moisture contents were calculated by using the following formula [33].

$$Moisture\ contents\ (\%) = \left[ \frac{Initial\ weight\ -\ Final\ weight}{Initial\ weight} \times 100 \right] \quad (2)$$

**Effect of pH.** The banana pulp was homogenized by using a blender and then sieved in a beaker through muslin cloth to determine the pH value at different stages of banana. A digital pH meter was used by dipping the sensory electrode in beaker after calibration [33].

**Titrable acidity (%).** Titrable acidity (TA) of banana pulp was measured by direct titration method in which 10 mL pulp was taken in beaker and 2–3 drops of phenolphthalein (1% solution) were added to it. For titration, NaOH (0.1 N solution in burette) was added dropwise in beaker until the color of sample turned light pink. Titrable acidity was calculated by using the following formula [33].

$$Titrable\ Acidity\ (\%) = \frac{Vol.NaOH\ (mL)X\ 0.1\ (Normality\ of\ NaoH)X0.064}{Pulp\ juice\ (mL)} \times 100 \quad (3)$$

**Measurement the starch content (%).** To study the starch conversion, anthrone test was conducted. For this reason, the 0.5g banana pulp was homogenized with 5ml 80% ethanol in air tight beaker and incubated in water bath for 30 minutes at 80°C. Then it was centrifuged at 44000 rpm for 5 minutes. After that, 20 mL of distilled water and 6.5mL of perchloric acid was added in it. Then centrifuged at 4°C for 20 minutes and the supernatant was saved as extract. This process was repeated second time by adding 5mL distilled water and the supernatant was added in first extract by making the volume upto 100 mL with distilled water. Then 0.1 mL was taken and final volume was made upto 1mL by adding distilled water. Absorbance was measured at 620 nm by single beam spectrophotometer. Further, stock solution of glucose was prepared by adding 100 mg glucose in 100 mL distilled water and standard solution was prepared by diluting the 10 mL of stock solution by making final volume upto 100 mL by using distilled water. The standard curve was drawn by using different concentration of standard solutions (0.1, 0.4, 0.6, 0.8, 1 mL) in each test tube and by making the volume of 1 mL. Then, 4 mL of anthrone reagent was added to each test tube and heated them for 8 minutes. After cooling, 5 mL distilled water was mixed with it. To measure the starch content, the value found for glucose was multiplied by 0.9 [34].

**Total soluble sugar content (%).** To access TSS, 15 g banana pulp from each treatment was blended with 45 mL distilled water and few drops of extract were placed on refractometer prism after centrifugation at 11000 rpm for 5 minutes. Brix reading (sugar contents in aqueous medium) was calculated by calibration of refractometer with distilled water [35].

**Measurement of phenolic content (mg GAE/ 100g) of banana.** Folin Ciocalteu (F.C.) colorimetric method was used to detect the phenolic content in banana. For this purpose, 0.5 mL banana extract was added in 0.5 mL F.C. reagent and it was homogenized manually for 20s. After 3 minutes, 7% sodium carbonate solution of 2 mL was added in control and experimental test tubes and were placed in boiling water for 1 minute. After cooling the mixture, absorbance was measured by single beam spectrophotometer at 760 nm. The results were presented in mg of Gallic acid equivalents (mg/10 g). A calibration curve was plotted by using different concentrations of standard solution of Gallic acid (20–100 mg/L) [36].

**Protein estimation (%).** Protein content in bananas was estimated by adding 0.25 g pulp extract in 10 mL of potassium phosphate buffer (50 mM) by adjusting its pH 7.8 at room temperature. After that, all samples were centrifuged for 15 minutes and 2mL Bradford reagent

was added in 0.1 mL aliquot of each sample. Absorbance of each sample was measured by spectrophotometer at 590 nm [37].

**Ethylene production (ppm).** To measure the ethylene concentration, all banana samples were weighed to record its initial weight and kept in the airtight plastic bag with rubber seal to prevent the gas leakage. Each bag contained 2–3 labelled banana samples. A three gas detector (F-950, Felix instruments, USA) with a gas sensor was used to measure the concentration of ethylene at different post-harvest stages of banana. Ethylene production was measured in each treatment of coated and controlled banana in triplicate [38].

## Statistical analysis

The experiment was carried out in a completely randomized design (CRD) in which each treatment was directed in three replications. In order to study the effect of each treatment, the data was subjected to ANOVA test at 5% level of significance and these statistical tests were subjected by using the "Minitab 19" software (Originated at Pennsylvania State University, USA). Means were separated by the Duncan's multiple range test. PCA (principal component analysis) was done to reduce multidimensional dataset.

## Results and discussion

### Synthesis and characterization of AgNPs

The formation of AgNPs from ELE indicated the visible change in color from colorless to dark brown. The UV-visible absorption spectra of AgNPs were recorded by mixing silver nitrate solution (2 mM) with eucalyptus leaf extract (Fig 1). AgNPs were characterized by using following analytical tools.

**UV-visible analysis of AgNPs.** UV-VIS absorption spectroscopy was used to evaluate the optical properties of AgNPs in which metal nanoparticles absorbed strong electromagnetic waves in the visible range and their formation showed the change in color from colorless to dark brown. The ELE and silver nitrate solution was mixed in different ratio for attaining the UV-visible absorption spectra of AgNPs. The results showed different peaks of surface plasmon resonance (SPR) in the range of 400–500 nm. The reaction mixture with 1:1 (v/v) ratio showed the highest absorbance peak of UV-VIS spectra at 419 nm while the mixture with 9:1 (v/v) did not show characteristic absorbance peak (Fig 2). The broad and lowest absorption peak at the lowest concentration of ELE was due to the formation of large anisotropic nanoparticles and low concentration of functional groups (alcohol, alkaloid and sugar) that were responsible for capping and stabilization of nanoparticles [39]. In general, absorbance peak intensity increased with an increase in plant extract and decreased with the decrease in plant extract. Moreover, absorbance peaks indicated a greater particle size at maximum wavelength and a smaller size at a shorter wavelength [27].

**FTIR analysis of AgNPs.** Fourier transform infrared spectroscopy (FTIR) analysis of AgNPs synthesized by using ELE is shown in Fig 3. It was performed to find out the functional groups which have an efficient role in capping, reduction and stabilization of AgNPs. The FTIR spectrum observed at 3361 cm$^{-1}$ correspond to N-H stretch containing alcohol or phenol and the band at 2915 cm$^1$ corresponds to C-H (alkane) stretch. Similarly, the bands found at 1606 cm$^{-1}$ associated with C = N, C = C and C = O stretching (Imine/alkene, conjugated alkene while 1441 cm$^{-1}$ contains C-H (alkene) respectively. The bands seen at 1358 cm$^{-1}$ attributes to C-H of the aldehyde group and peak at 1225 showed the C-O stretching of ester. Another peak at 1030 cm$^{-1}$ and 833 cm$^{-1}$ indicated that C-O, N-H and C-H functional group might also be bounded with AgNPs respectively. So the result revealed that these functional groups were responsible for the formation and stabilization of AgNPs as a capping agent due to the

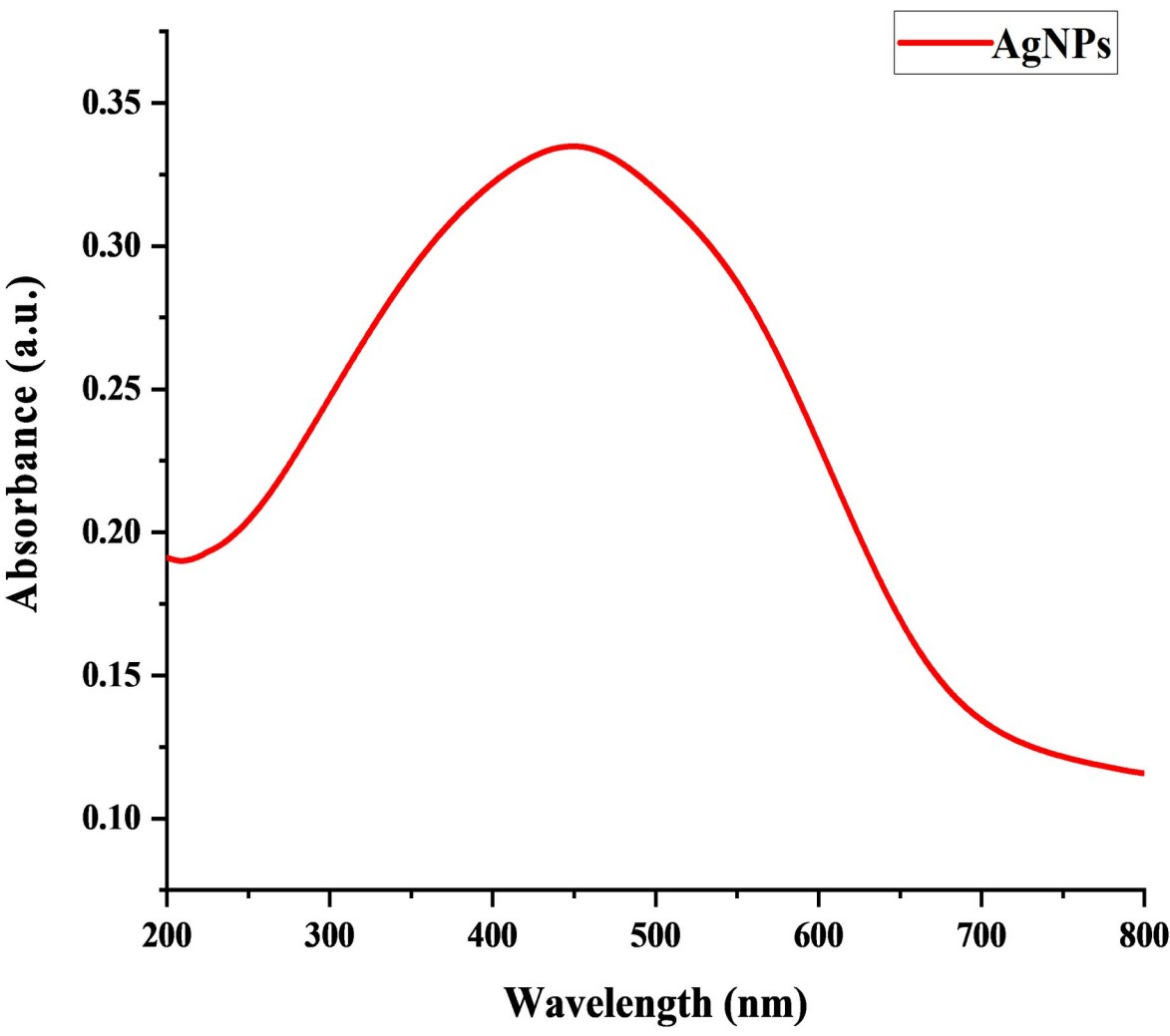

**Fig 1. Uv-vis spectrum of green synthesized AgNPs from ELE.**

presence of soluble elements in ELE containing high percentage of C-N, C-C and O-H molecules [28,40]. Several reports have proposed that the phytochemicals present in ELE as stabilizing and capping agent play a crucial role in the reduction of silver ions [27,41,42].

**XRD analysis of AgNPs.** X-ray diffraction analysis is used to study the structure and crystal size of nanoparticles. XRD pattern of AgNPs showed that main peaks were 38.1˚,44.2˚,64.5˚ and 77.3˚ at 2 theta that can be corresponded to the (111), (200), (220) and (311) crystallographic planes (JPDS Card No. 04-0783). In addition, two other unassigned weak peaks appeared at 27.8˚ and 32.1˚ at 2 theta that might be due to the presence of organic compounds in ELE (Fig 4). The average size of AgNPs was 10 nm as calculated by estimating (Full Width Half Maximum) FWHM of peak (111) through Debye–Scherrer's equation [43]. The sharp peak in XRD pattern confirmed the existence of capping agent in leaf extract that stabilized the nanoparticles. Similarly, weaker peaks might be due to the presence of some biological macromolecules. In XRD analysis, height, width and position of peaks illustrated the size of AgNPs. The increase in peak showed the smaller size of nanoparticles. Similar results were reported by researcher who find the peak at 38.45˚, 44.67˚, 64˚ and 77˚ that were due to the presence of phytochemical compounds in extract [42].

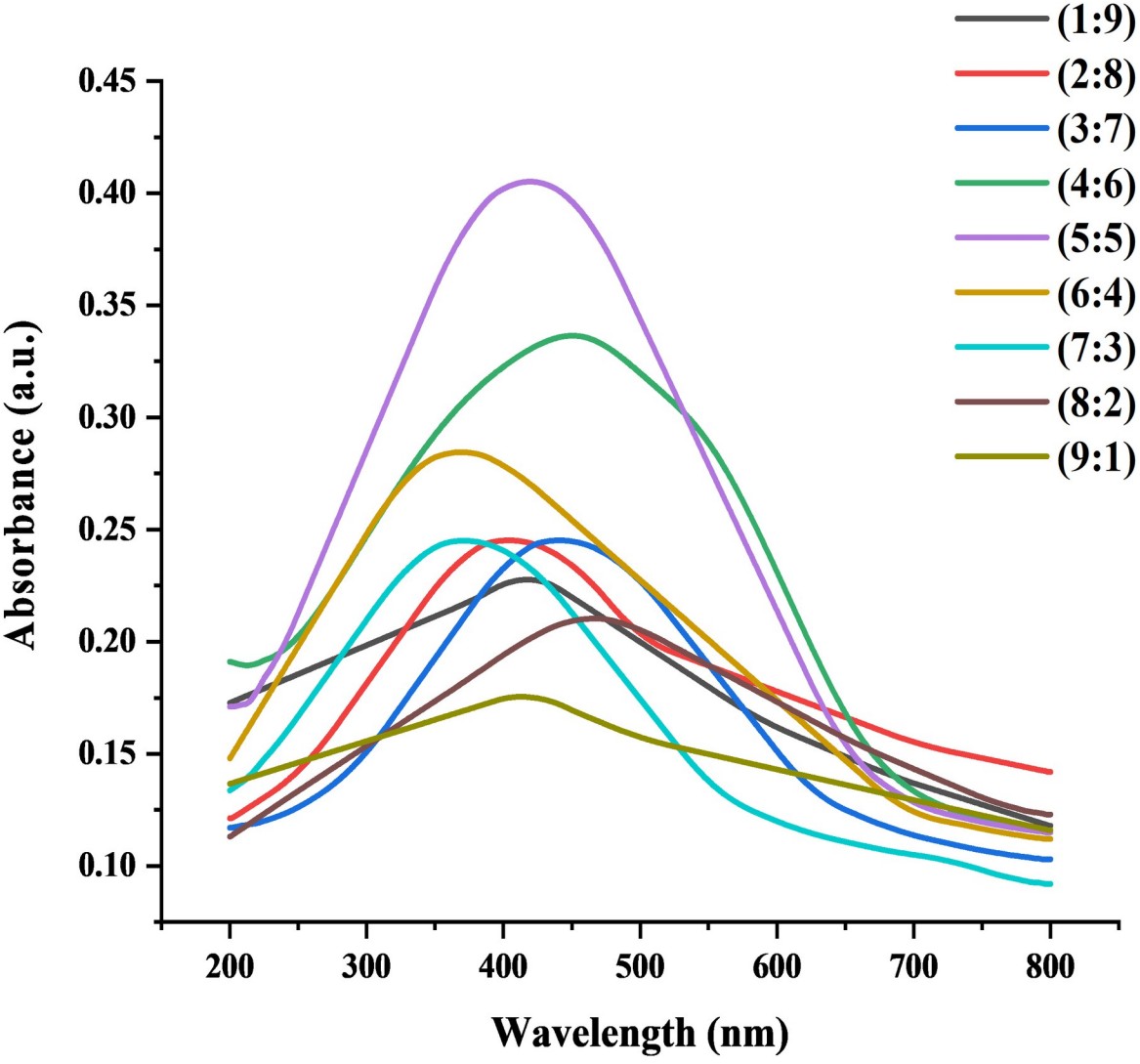

**Fig 2. Optimization of AgNPs with different ratio of silver nitrate and ELE solution.**

Similarly, the results of AgNPs synthesis were reported where the highest peak were recorded at range of 38.0–38.9 [31,43–45].

**SEM analysis of AgNPs.** Scanning electron microscope is used to analyze the surface morphology, size and shape of nanoparticles at different magnification. The results depicted the spherical agglomerated particles with the range of 50–100 nm in a micrograph of AgNPs and the scale was used at range of 1–10 μm (Figs 5 and 6). The results of the present work are parallel with previous studies that reported the spherical AgNPs synthesis by pomegranate peel extract [29], seed extract of dates [46] and extract of cheese weed mallow [47].

**TEM analysis of AgNPs.** Transmission electron microscopy as imaging tool confirmed the microstructure, morphology and size distribution of AgNPs that were synthesized from ELE. The experiment revealed the spherical shape and ploy dispersal nature of these particles as well as its crystalline structure with size range of 5–20 nm (Figs 7 and 8). The close observation of AgNPs showed the shaded layer surrounding them and these foreign components come from leaf extract act as capping agent used to stabilize AgNPs [48]. Earlier

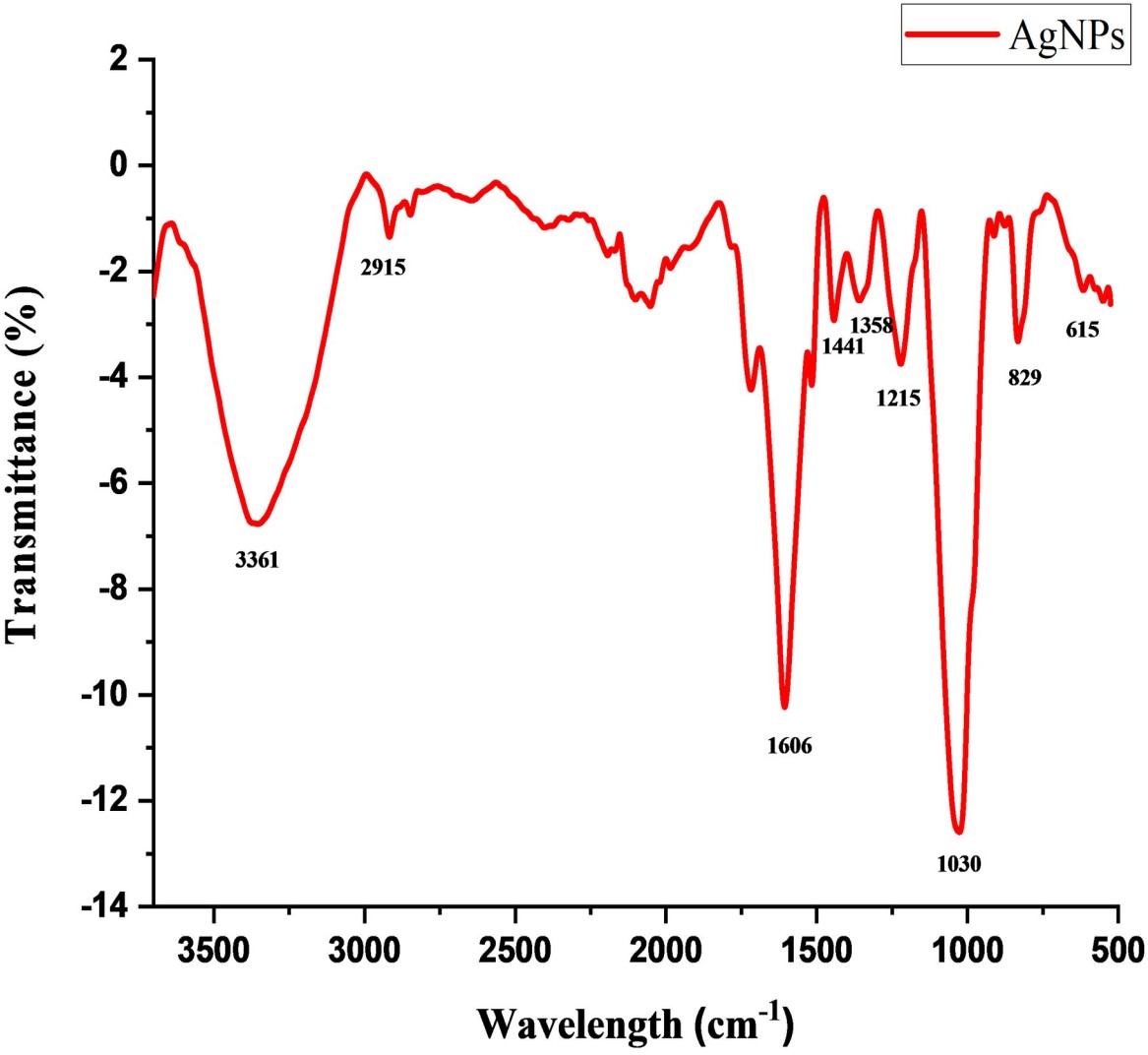

**Fig 3. FTIR analysis of green synthesized AgNPs.**

investigations also reported the spheroidal morphological structure with stabilizing agents [30,49,50].

**EDX analysis of AgNPs.** Energy-dispersive X-ray spectroscopy (EDX) was used to reveal the elemental composition of the AgNPs. The analysis reported the strong peak of silver at 3 keV with standard weight percent that was 70% and weak signals of other organic compounds as carbon, oxygen and chlorine at range of 0–2.5 keV (Fig 9). The highest peak indicated the presence of silver element while the minor peak revealed the presence of other soluble elements in leaf extracts that act as stabilizing agents of AgNPs [51]. Previous research work stated the quantitative information of AgNPs solution which confirmed the 72% silver component by using onion extract, 69% silver by tomato extract and 77% silver medicinal plant (*Carduus crispus*) extract [31,42].

## Application of AgNPs on banana

Different concentrations of AgNPs were applied on banana samples to select the best concentration of nanoparticles involved in reducing the post-harvest losses in banana.

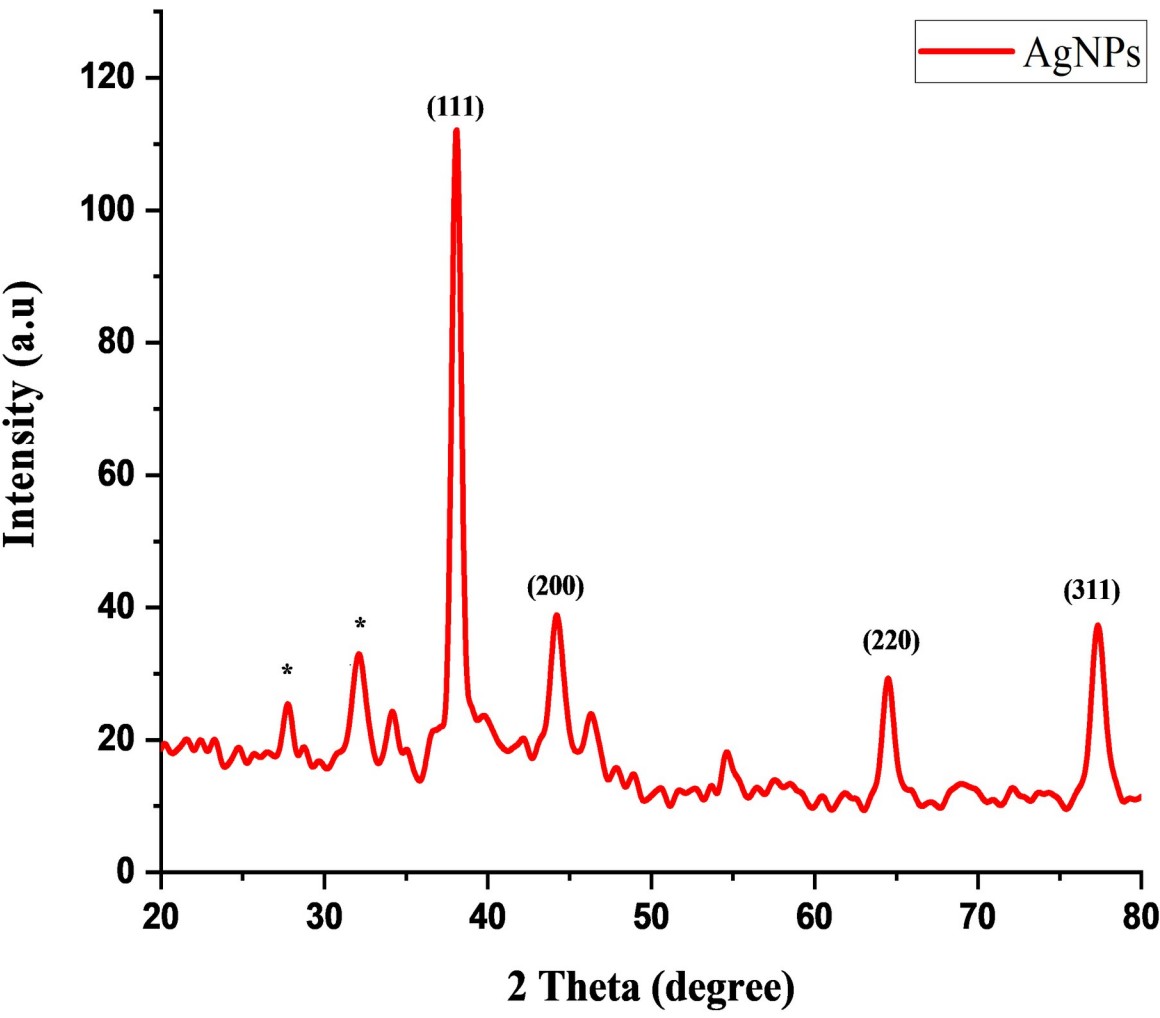

**Fig 4. XRD analysis of green synthesized AgNPs.**

**Effect of AgNPs on banana color.** Color is the most important criterion to determine the quality of fresh fruit. The surface color of banana, which changes from green to yellow as the fruit ripens, is one of the key elements to study the quality and market value of fruit [38]. All the samples of banana treated with different levels of AgNPs (T1 = 0.01, T2 = 0.02, T3 = 0.03, T4 = 0.04 and T5 = 0.05%) resulted the least change in banana color as compared to control banana samples. The untreated banana samples reached to fully ripened stage within 14 days of storage period and it may be due to degradation of green pigment chlorophyll that is replaced by new pigments like carotenoids, microbial growth and more ripening due to ethylene production [52] and the samples treated with AgNPs reduced the postharvest losses till 32 days at 25°C. Among all concentrations, bananas treated with 0.01% (T1) of AgNPs showed the lowest score (3.00) of peel color with more greenish shade at 32 days of storage while untreated bananas got the highest score of 7.00 at 21 days and score 8.00 at 28 days of storage (Fig 10). The results are consistent with previous reports that AgNPs from citrus peel extract delayed the color development in different fruits and vegetables at 0.1% concentration of neem-AgNPs that delayed the color of banana by minimizing the microbial attack [53]. Similarly, AgNPs from marigold flower and lemon peel extract improved the shelf life of berries

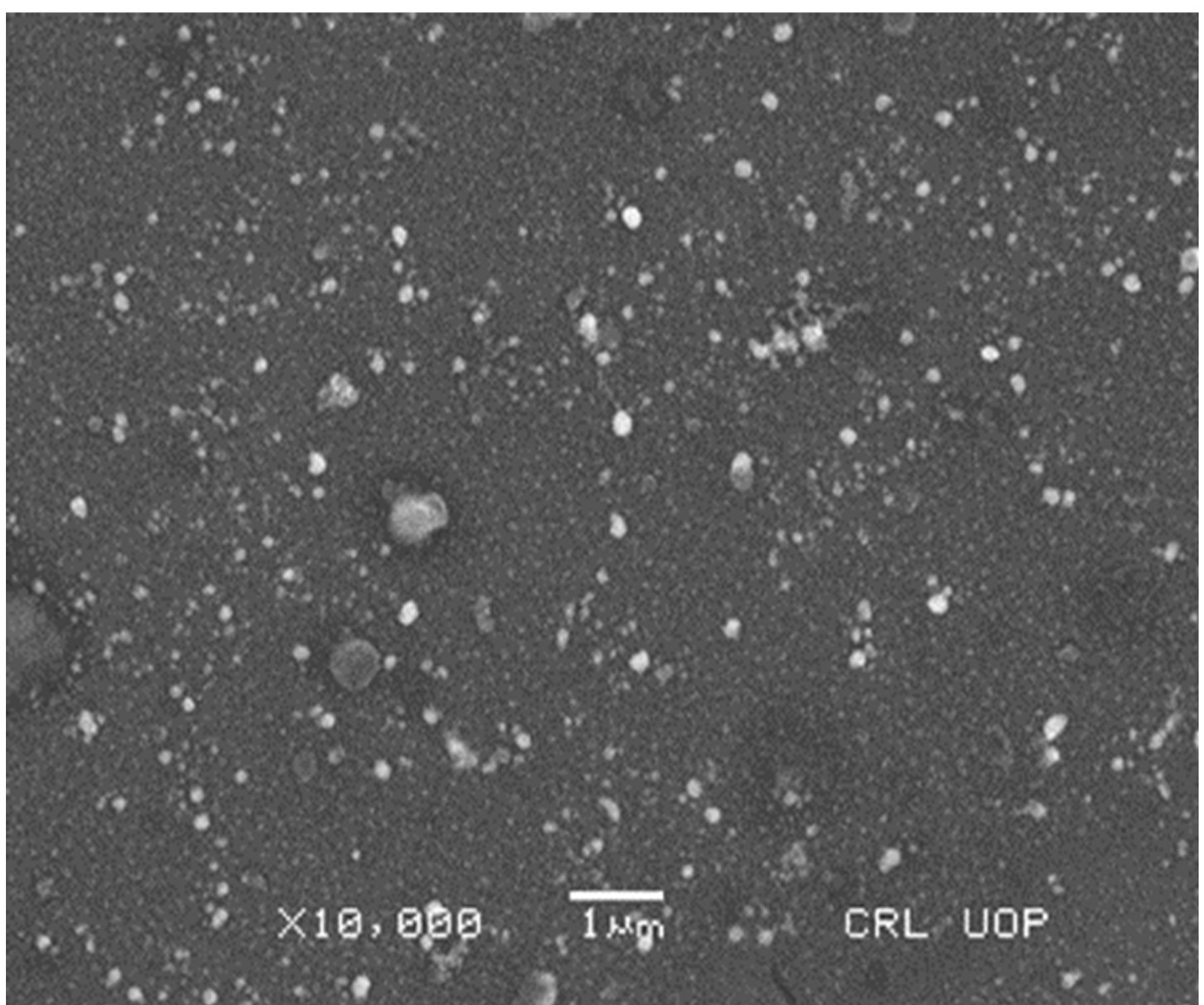

**Fig 5. SEM image of green synthesized AgNPs with resolution power of X 10,000.**

upto 7 days [54] as these nanoparticles have the ability to reduce ethylene production and control microbial growth that inhibit the degradation of chlorophyll and formation of carotenoid in banana peel and retain the banana color fresh [55,56].

**Effect of AgNPs on banana decay.** Banana fruit decay was observed primarily after harvesting, followed by marketing, transportation and storage. The banana samples treated with different concentrations of AgNPs showed 0 decay during early storing stage. It was noted that the untreated bananas showed maximum decay at 14 days as compared to other treatments. Further, the treated samples showed maximum increase in shelf life with 0.01% and 0.02% AgNPs and got the value of 3.6 and 5.6 respectively for decay at 32 days of storage time. As compared to 0.01% and 0.02% of AgNPs, other three treatments as 0.03%, 0.04% and 0.05% concentration of AgNPs indicated maximum decay that was 7.6, 7.6 and 8.0 respectively at 32

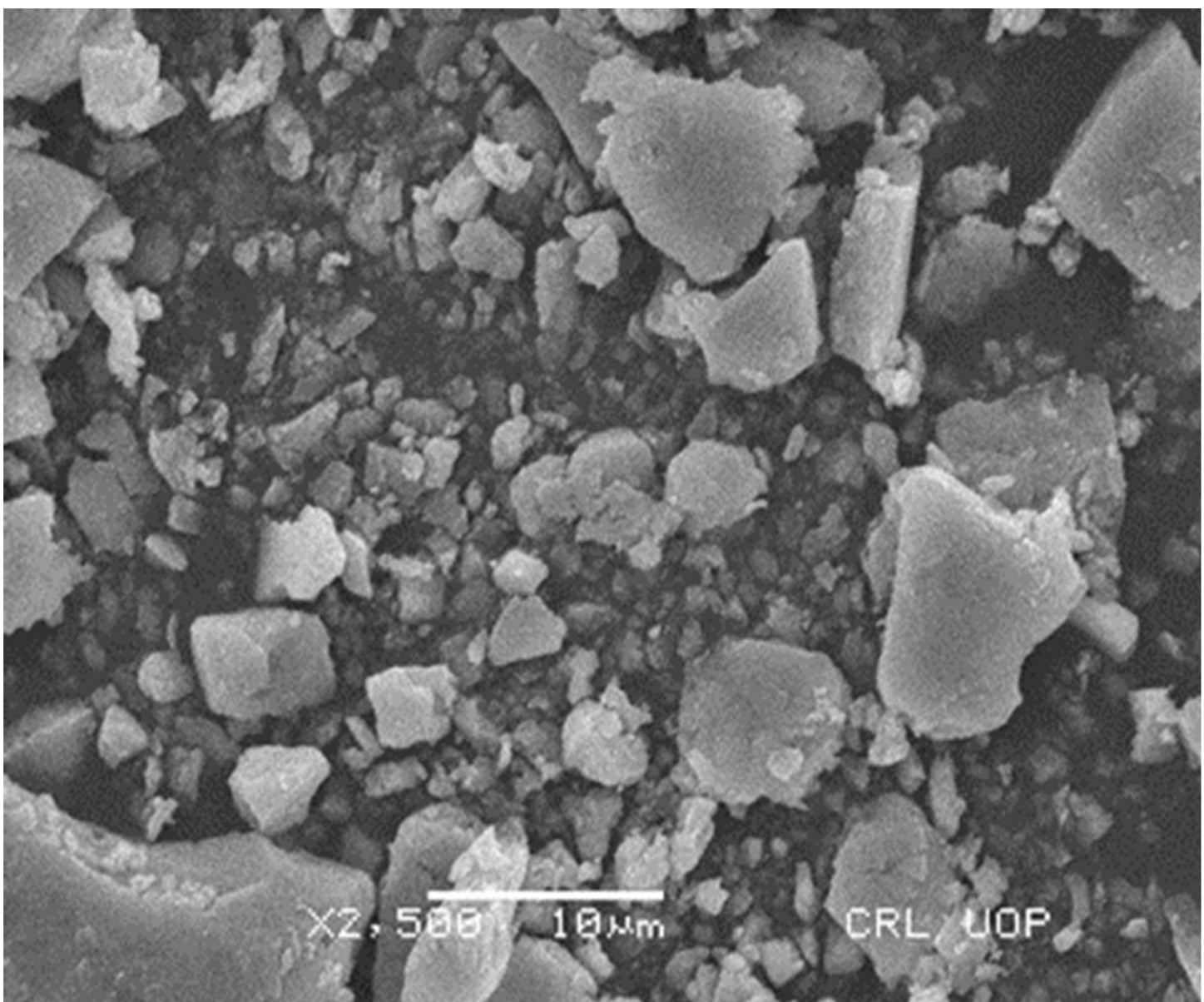

**Fig 6. SEM image of green synthesized AgNPs with resolution power of X 25,00.**

days of storage. So, the results revealed that 0.01% and 0.02% were responsible to delay the process of decay in banana except control that showed full ripeness within 14 days. Within 21 days of storage, bananas became completely non edible with the maximum score of 8.0 (Fig 11) that may be due to the fast fruit ripening as a result of higher ethylene production and microbial attack that was the key factor to fruit deterioration. Further, fresh fruits continues to respire and transpire after harvesting that ultimately extend the decay rate in such fruits [57]. Generally, the results were in alignment with previous research work in which AgNPs attained from marigold flower and lemon peel extract increased the shelf life of berries for 7 days [54] and AgNPS with maize starch controlled the post-harvest decay in apricot for 8 days at 25˚C and 24 days at 6˚C [58]. Silver nanoparticle treatment maintained water loss and retarded the microbial growth in fruits by controlling the permeability of membranous tissues for

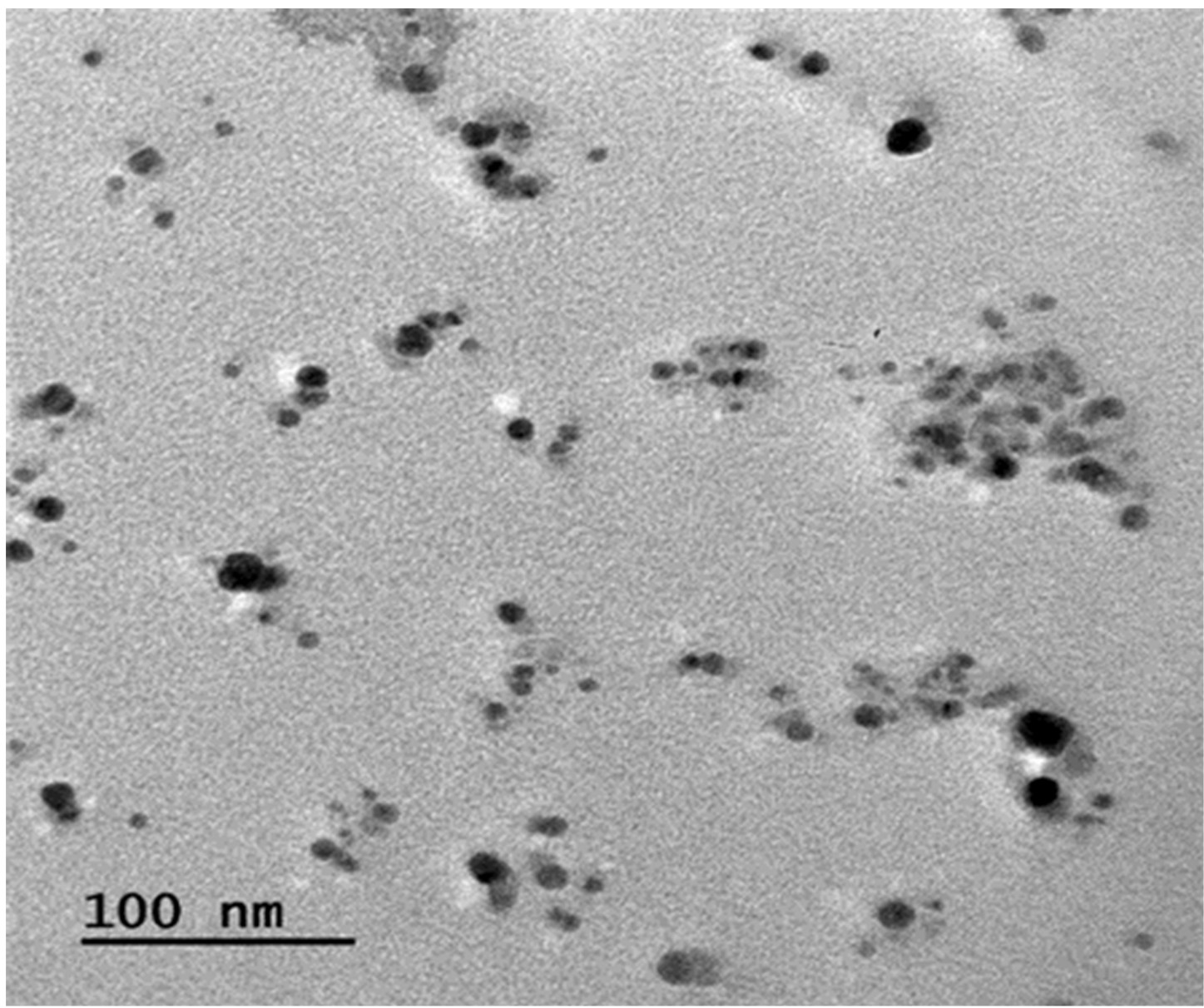

**Fig 7. TEM image of green synthesized AgNPs from ELE.**

minimization of decay process as well as inhibition of ethylene production showed the shelf life enhancement in different fruit cultivars [30,55].

**Effect of AgNPs on banana firmness (N).** The firmness of all banana samples was in range of 6.2–7.1 at 14 days of storage after application of AgNPs with different concentrations including untreated bananas. After 14 days of storage, bananas showed a decrease in firmness with control (3.3) while treated banana showed less decrease in firmness that was 6.2, 5.7, 5.5, 4.8 and 4.8 with 0.01%, 0.02%, 0.03% 0.04% and 0.05% AgNPs respectively. In control, firmness loss was noted as 0.9 and it showed complete decay at day 21 and this might have contributed for the textural changes of banana due to change in amount of starch and pectic substances and polysaccharides in banana pulp [52]. The less increase in firmness as compared to control was observed 4.3, 4.1, 3.2, 2.4 and 2.1 with 0.01%, 0.02%, 0.03%, 0.04% and 0.05% of

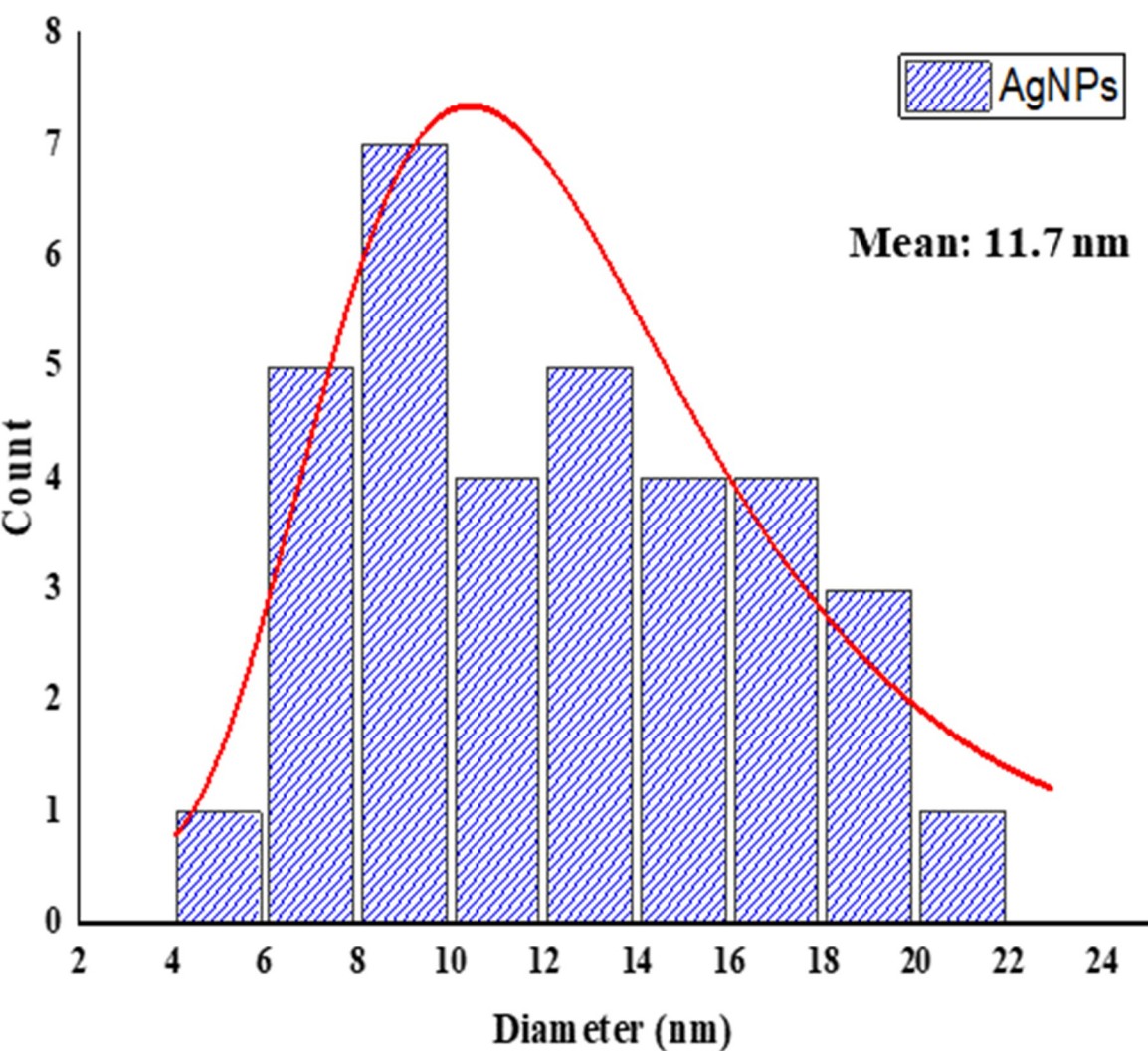

**Fig 8. TEM size distribution histogram of AgNPs.**

AgNPs respectively till 32 days of storage period (Fig 12). The results are in accordance with the study that used the guar gum based AgNPs with carboxymethyl cellulose to enhance the shelf life of mango for 14 days at 25˚C after the spoilage of untreated mango over the total storage period of 28 days [59]. The loss of firmness during storage period was due to the change in banana texture but the coating of silver nano particles slowed down the metabolic and enzymatic activities by inhibiting the respiratory metabolism in the fruits that results a slower degradation of pulp tissues [59,60]. Consequently, bananas stored with AgNPs can retain their firmness by reducing the ethylene production, respiration rate, the speed of converting sugar and decay rate of banana [61].

**Effect of AgNPs on weight loss (%) of banana.** Weight loss is an important element to determine the quality of banana during prolonged storage period and to investigate the increase in the shelf life of banana. The results showed the weight loss percentage range of cavendish banana was 3.2–4.3% during storage time of 7 days that depends on the size of banana. During 14 days of storage, the weight loss percentage slightly increased in all banana samples at the range of 5.8–6.6% except control that was 9.3%. Similarly, the weight loss percentage

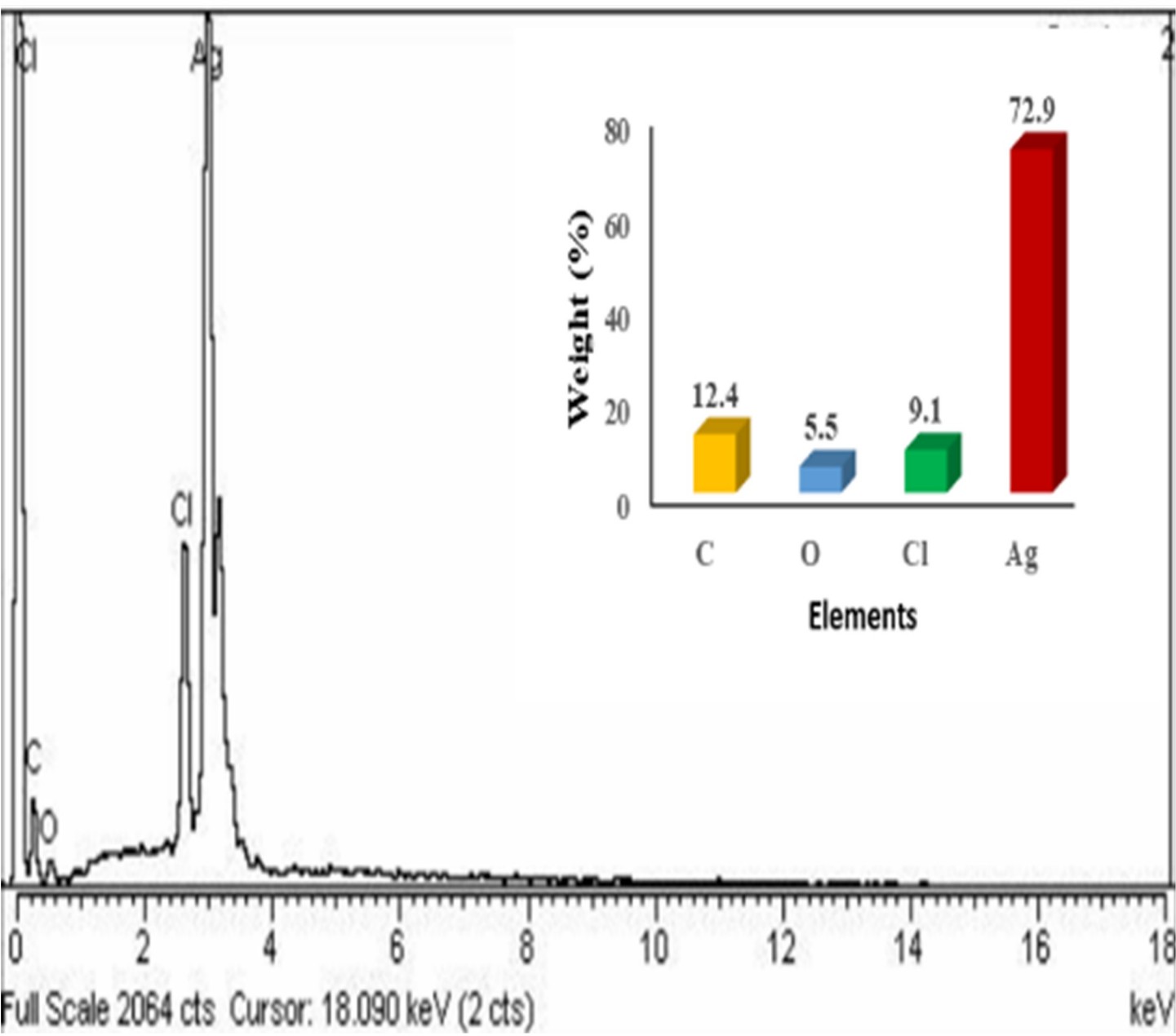

**Fig 9. EDX analysis of green synthesized AgNPs.**

was 13.9% in control banana after 21 days, 21.5% at 28 days and 33% at 32 days of storage and they became non-edible. This could be linked to the reduced moisture retaining ability of un treated banana due to accelerated rate of transpiration and respiration from banana surface as well as deterioration in tissues of banana peel [62], while the weight loss percentage showed less increase as 10.8%, 12.8%, 12.8%, 11.6% and 16.8% with 0.01%, 0.02%, 0.03%, 0.04% and 0.05% AgNPs respectively at 32 days of storage (Fig 13). The results are similar to the previous work where AgNPs synthesized from tea extract increased the weight loss percentage with increase in storage period in cherry tomatoes for 15 days [63]. The increase in weight loss per-centage with storage time was due to AgNPs coatings that act as semipermeable barriers which

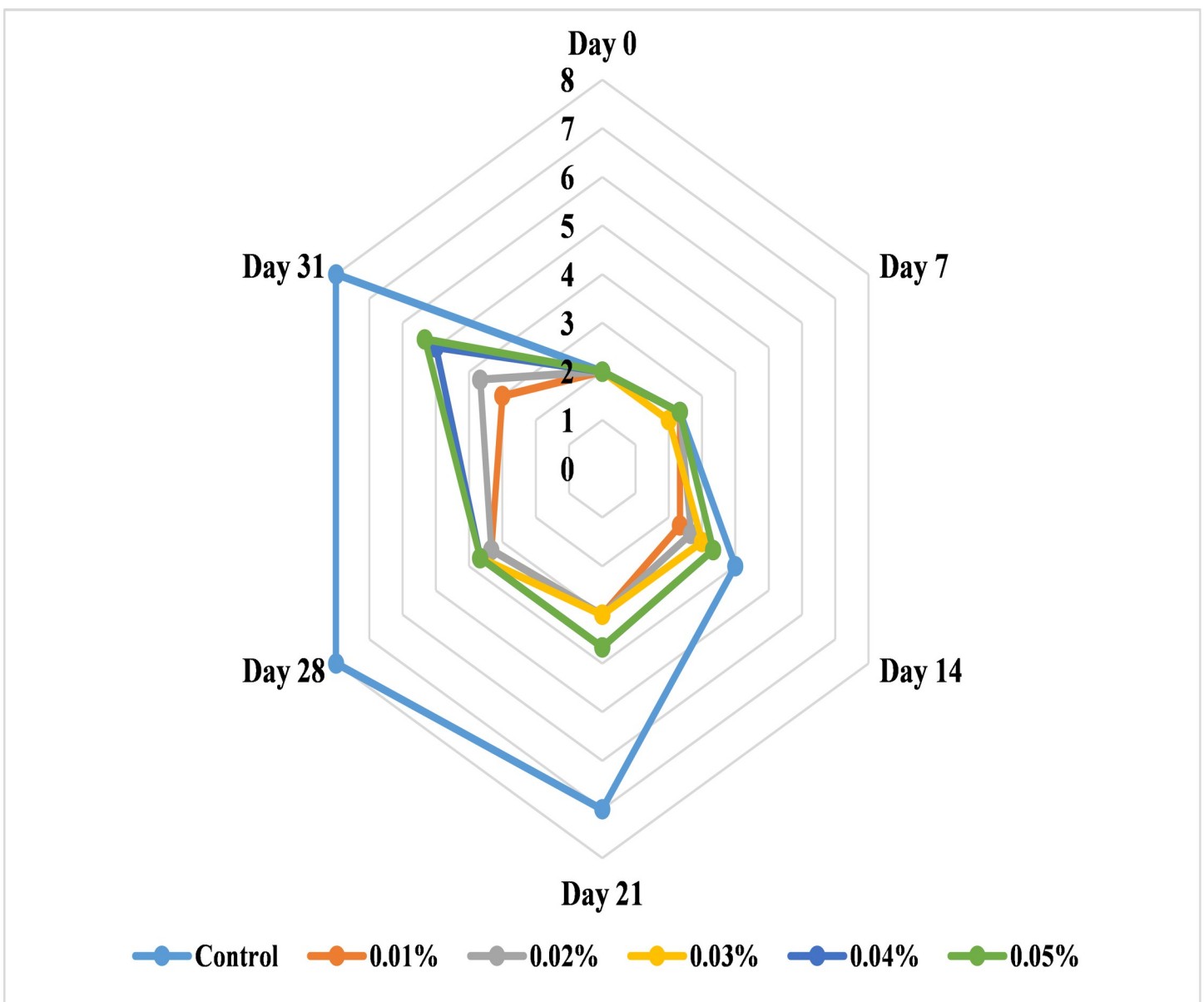

**Fig 10. Effect of AgNPs concentrations (T0 = control, T1 = 0.01%, T2 = 0.02%, T3 = 0.03%, T4 = 0.04%, T5 = 0.05%) on color of Cavendish banana (BASRAI) during 32 days of storage.**

hinders the oxygen, carbon dioxide, water loss, respiration and other oxidation reactions and maintain the weight loss percentage in fruits by improving their post-harvest quality [6,58].

**Effect of AgNPs on pulp to peel ratio (%) of banana.** Pulp to peel ratio is a consistent index to study the post-harvest losses during ripening of banana, that also reveal the change in their moisture contents [64]. The effect of various AgNPs coatings on pulp to peel ratio of bananas showed the range of 1.2–1.3% with all treatments at the start of experiment and the treated banana with 0.01%, 0.02% and 0.03% had more effect on percentage of pulp to peel ratio as compared to 0.04% and 0.05%. Basrai showed less increase in pulp to peel ratio percentage that was 1.4–1.6% at day 7 and 1.9–2.4% at day 32. As clearly observed in (Fig 14) the pulp to peel ratio of uncoated banana increased rapidly that was 2.4% on day 14 and 3.4% on

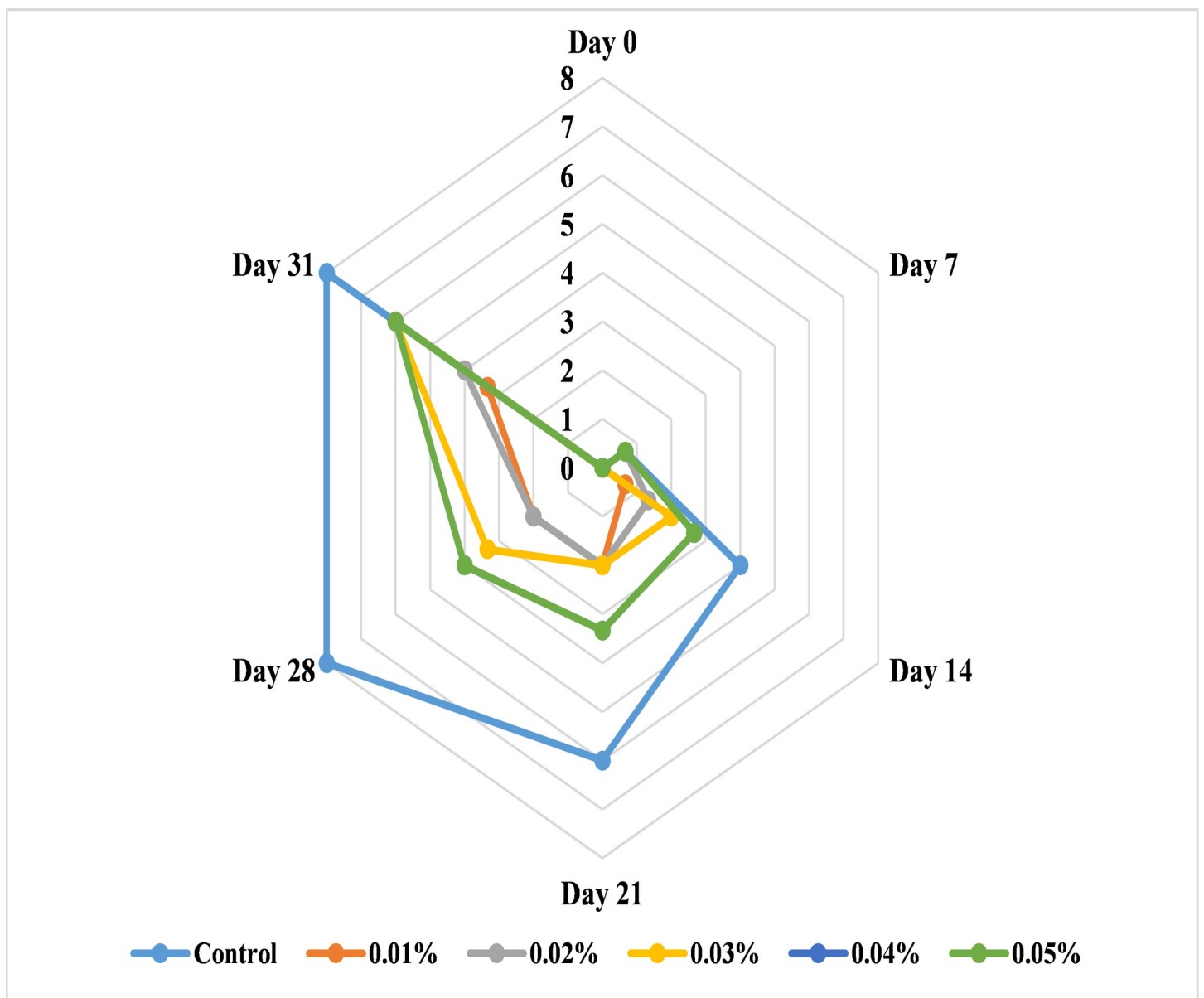

**Fig 11. Effect of AgNPs concentrations (T0 = control, T1 = 0.01%, T2 = 0.02%, T3 = 0.03%, T4 = 0.04%, T5 = 0.05%) on decay of Cavendish banana (BASRAI) during 32 days of storage.**

day 32 with control. So, it was noted that when the peel weight was divided by the pulp weight in the current study, the pulp to peel ratio of bananas showed more upward tendency in untreated banana as compared to treated bananas. This might be due to water loss from peel to pulp and atmosphere as well as the increased amount of soluble sugar in pulp ultimately increased the pulp to peel ratio by transferring the osmotic pressure in the pulp more quickly [65]. Similarly, the pulp to peel ratio was 1.6% with 0.01%, 0.02%, 0.03%, 0.04% and 1.7% with 0.05% AgNPs on day 14. It was revealed that all banana samples with AgNPs treatments got less increase in pulp to peel ratio at 32 days of storage in which 0.03% (2.1%), 0.04% (2.1%) and 0.05% (2.5%) that was less adequate for post-harvest quality of banana as compared to 0.01% (1.8%) and 0.02% (1.9%) with more appropriate results (Fig 7A). The results are similar

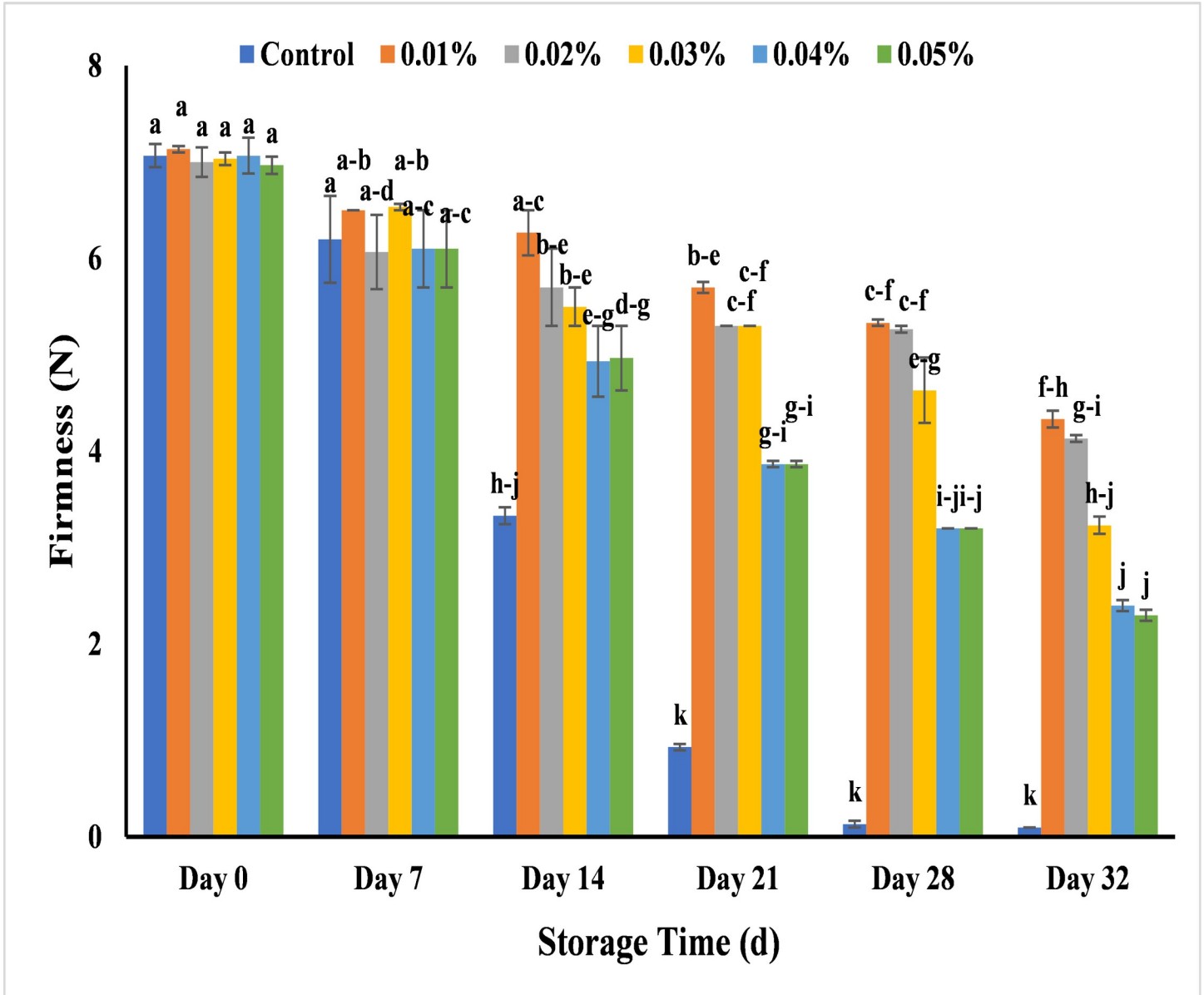

**Fig 12. Effect of AgNPs concentrations (T0 = control, T1 = 0.01%, T2 = 0.02%, T3 = 0.03%, T4 = 0.04%, T5 = 0.05%) on firmness of Cavendish banana (BASRAI) during 32 days of storage.**

to the work in which 1.15% and 1.25% concentration of chitosan NPs increased the shelf life of banana upto 11 days with higher pulp to peel ratio in un treated banana as compared to treated banana samples [8]. So, the less increase in pulp to peel ratio was due to AgNPs coatings that reduce the respiration and other oxidation reactions and maintain the pulp to peel ratio in fruits by improving their post-harvest quality. The rapid conversion of starch into sugar in banana pulp during ripening process and the created osmotic gradient increases the weight of pulp while transpiration and respiration loss from peel again decrease the weight of peel [33]. AgNPs act as barrier and maintain the moisture evaporation and respiration of gases and tran- spiration rate from coated banana surface. Ultimately, treated banana fruits showed less increase in pulp to peel ratio than uncoated fruits [6,58].

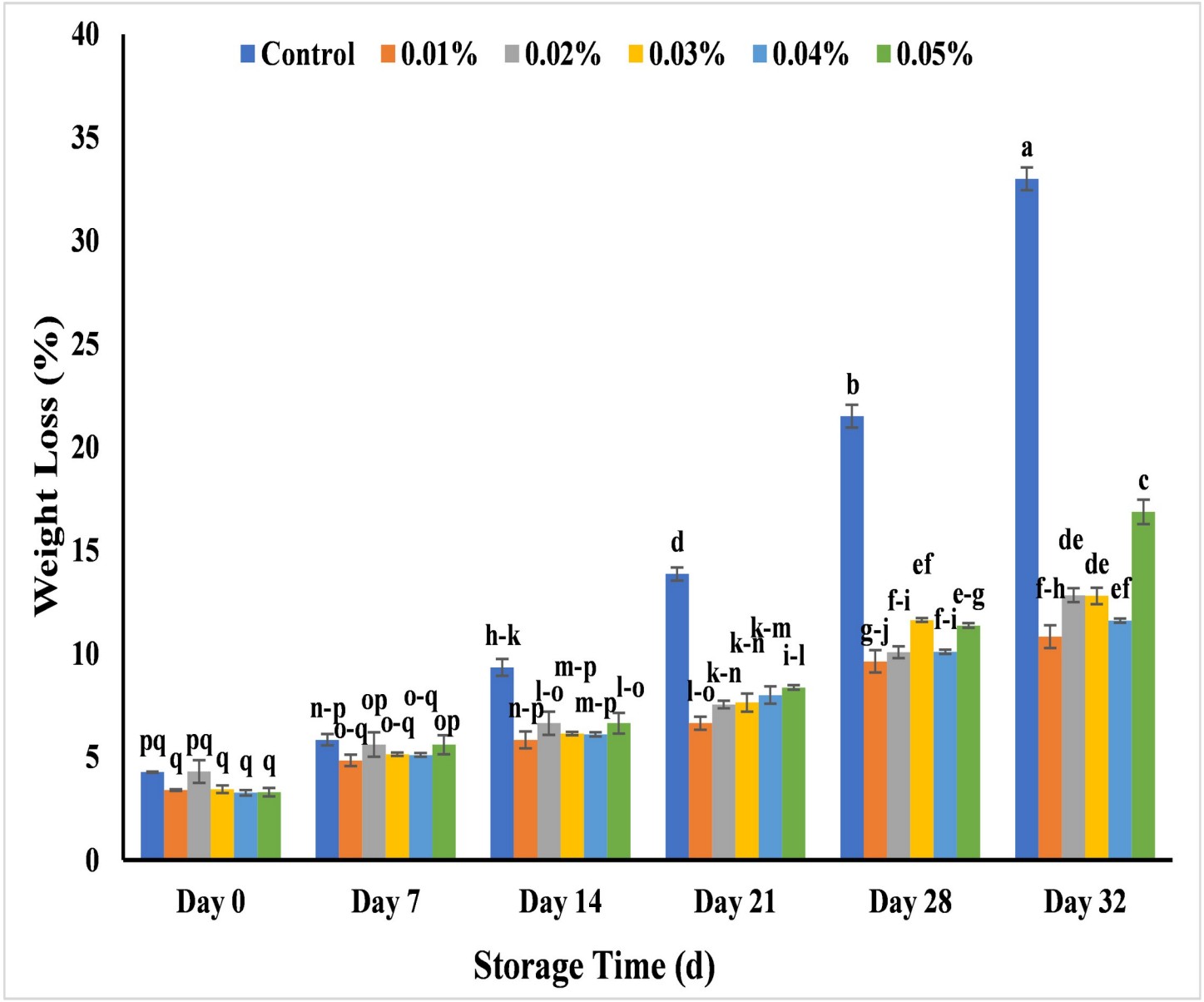

**Fig 13. Effect of AgNPs concentrations (T0 = control, T1 = 0.01%, T2 = 0.02%, T3 = 0.03%, T4 = 0.04%, T5 = 0.05%) on weight loss percentage of Cavendish banana (BASRAI) during 32 days of storage.**

**Effect of AgNPs on moisture content (%) of banana.** The study of moisture content is another trend to check the post-harvest quality of banana. It was recorded that moisture content was in range of 77.6–82.0 including control banana at 7 days of storage, it gradually decreased and attained the value of 73.3% in untreated banana. The value of moisture content of untreated banana reached to 56% on day 21 and completely destroyed at day 28 with reduced moisture content. Rather than control, treated banana samples showed less increase in moisture content that was 66.3%, 63%, 60.6%, 60.6% and 59.6% with 0.01%, 0.02% and 0.03% 0.04% and 0.05% AgNPs respectively at the end of storage (32 days). This might be due to more respiration rate and evaporation in untreated banana than treated bananas [66]. The banana samples exhibited minimum decay and decrease in moisture content with 0.01%,

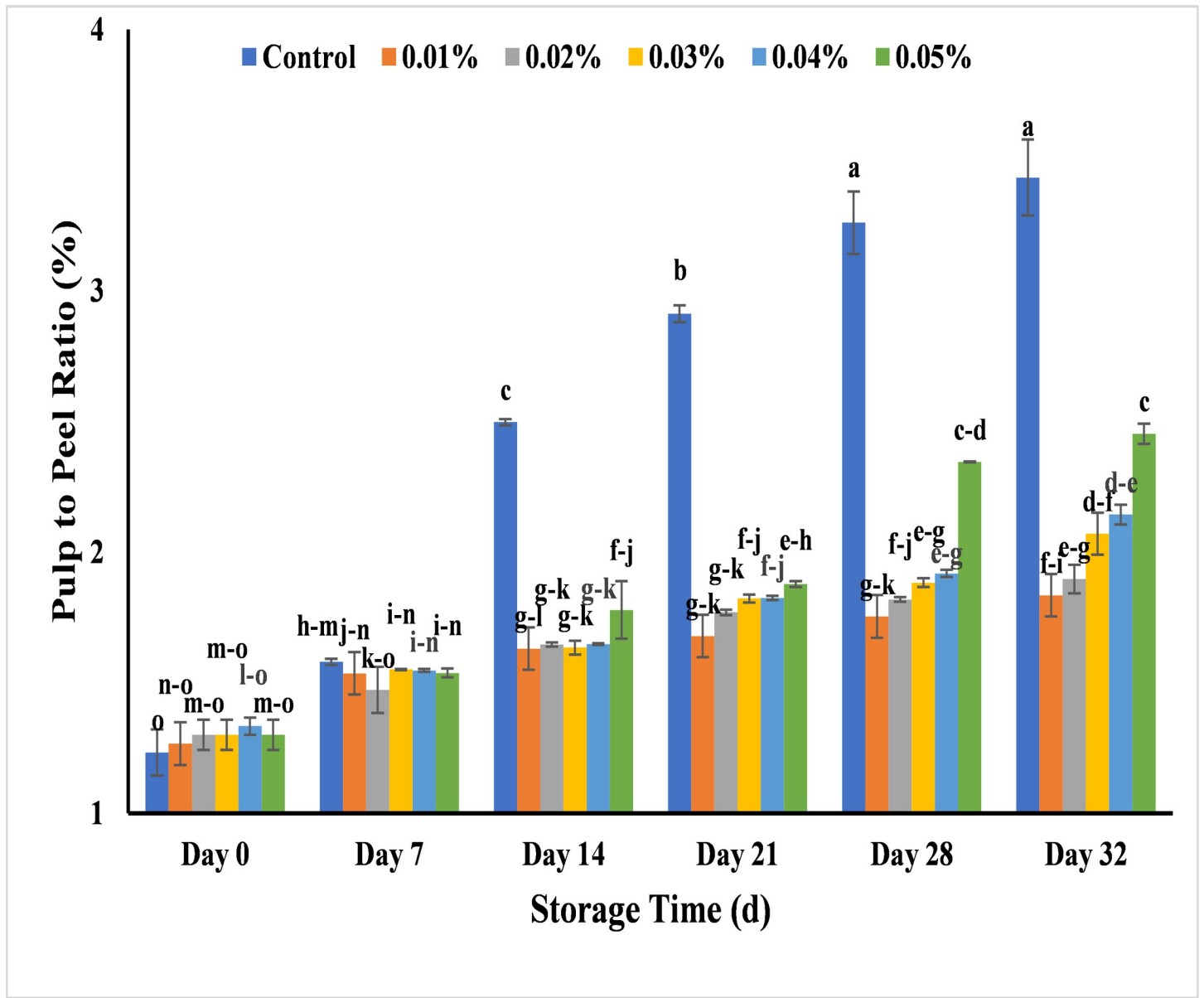

**Fig 14. Effect of AgNPs concentrations (T0 = control, T1 = 0.01%, T2 = 0.02%, T3 = 0.03%, T4 = 0.04%, T5 = 0.05%) on pulp to peel ratio (%) of Cavendish banana (BASRAI) during 32 days of storage.**

0.02% and 0.03% as compared to 0.04% and 0.05% of AgNPs (Fig 15). Similar trends were obtained by researchers who used 0.01% AgNPs with Polyvinyl pyrolidone (PVP) solution and used these nanoparticle coatings in packaging of Rutab dates to enhance their shelf life for 30 days at 12°C in Italy [67] and coating of AgNPs to maintain the post-harvest quality in strawberry by increasing its shelf life for 16 days [68]. The researchers explained the role of AgNPs in reduction of respiration or evaporation that increased the shelf life of treated banana samples by maintaining the moisture level in fruits and vegetables [63,67].

**Effect of AgNPs on pH of banana.** Banana pulp pH was influenced by different treatments of AgNPs. Normally, it decreases at the start of the ripening stage and continuously increase until it attains a fully ripen stage [69]. Untreated banana showed pH values in range

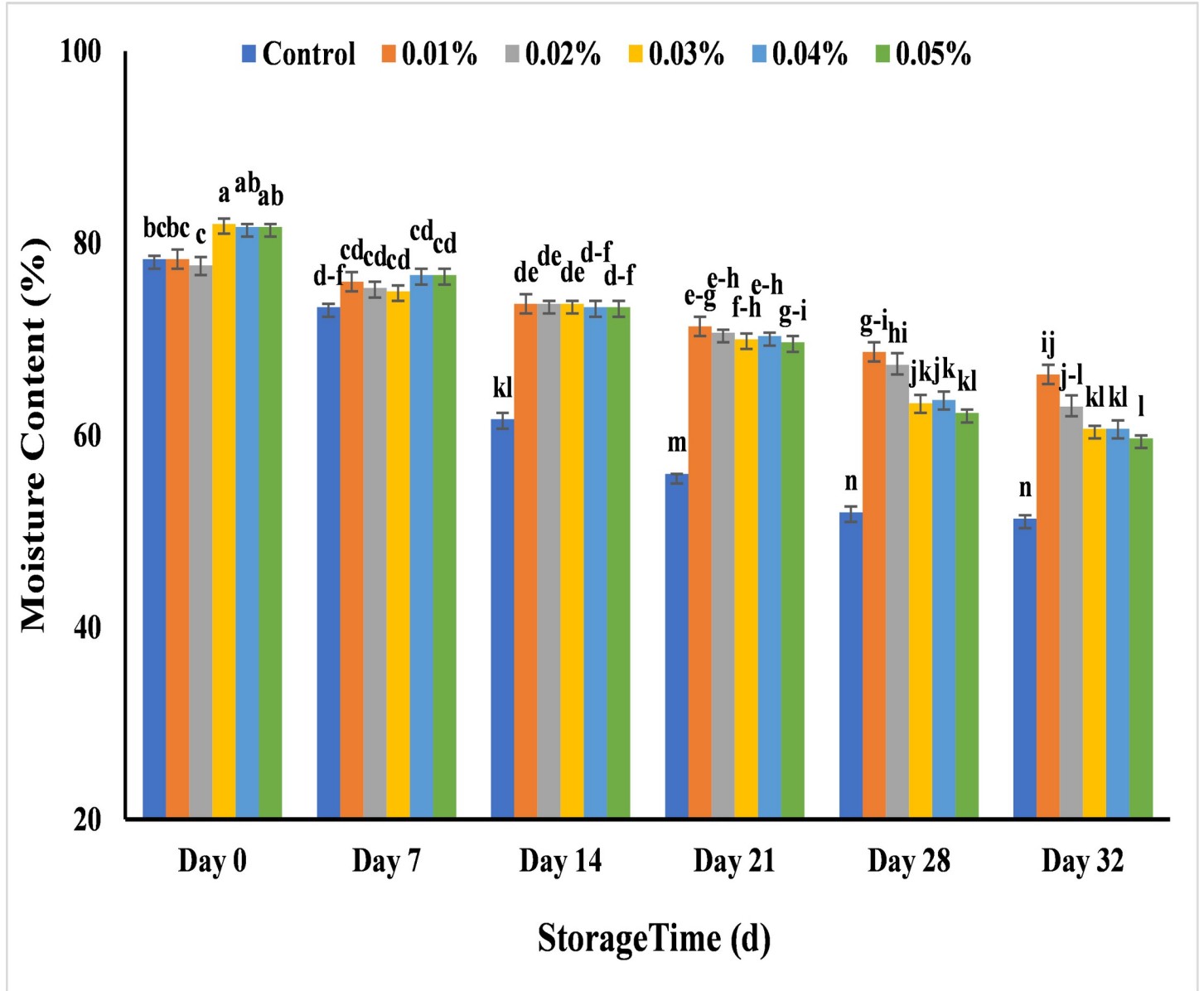

**Fig 15. Effect of AgNPs concentrations (T0 = control, T1 = 0.01%, T2 = 0.02%, T3 = 0.03%, T4 = 0.04%, T5 = 0.05%) on moisture content (%) of Cavendish banana (BASRAI) during 32 days of storage.**

of 5.3–5.6 along with all banana samples treated with different concentrations of AgNPs at stage 0 to 7. It was noted that the control banana sample of Basrai started decay at pH 6.0 on day 21 and showed complete decay at pH 6.3 on day 28. The sharp increase in pH was due to change in metabolic process and less acidity that is directly proportional to respiration rate in fruit [70]. In comparison to control, all treated banana samples maintain pH till 32 days of storage with slight change. Treated banana depicted the less change in pH as 5.6 with 0.01% and 5.5 with 0.02% among all treatments at 32 days of storage (Fig 16). The results showed similarities with previous literature in which AgNPs maintain the pH value during the 30 days storage period of loquat at 4°C [70] and 10 days storage period of carrot at 10°C [71] while the sharp increase was seen in control samples and the fluctuations in results could be due to

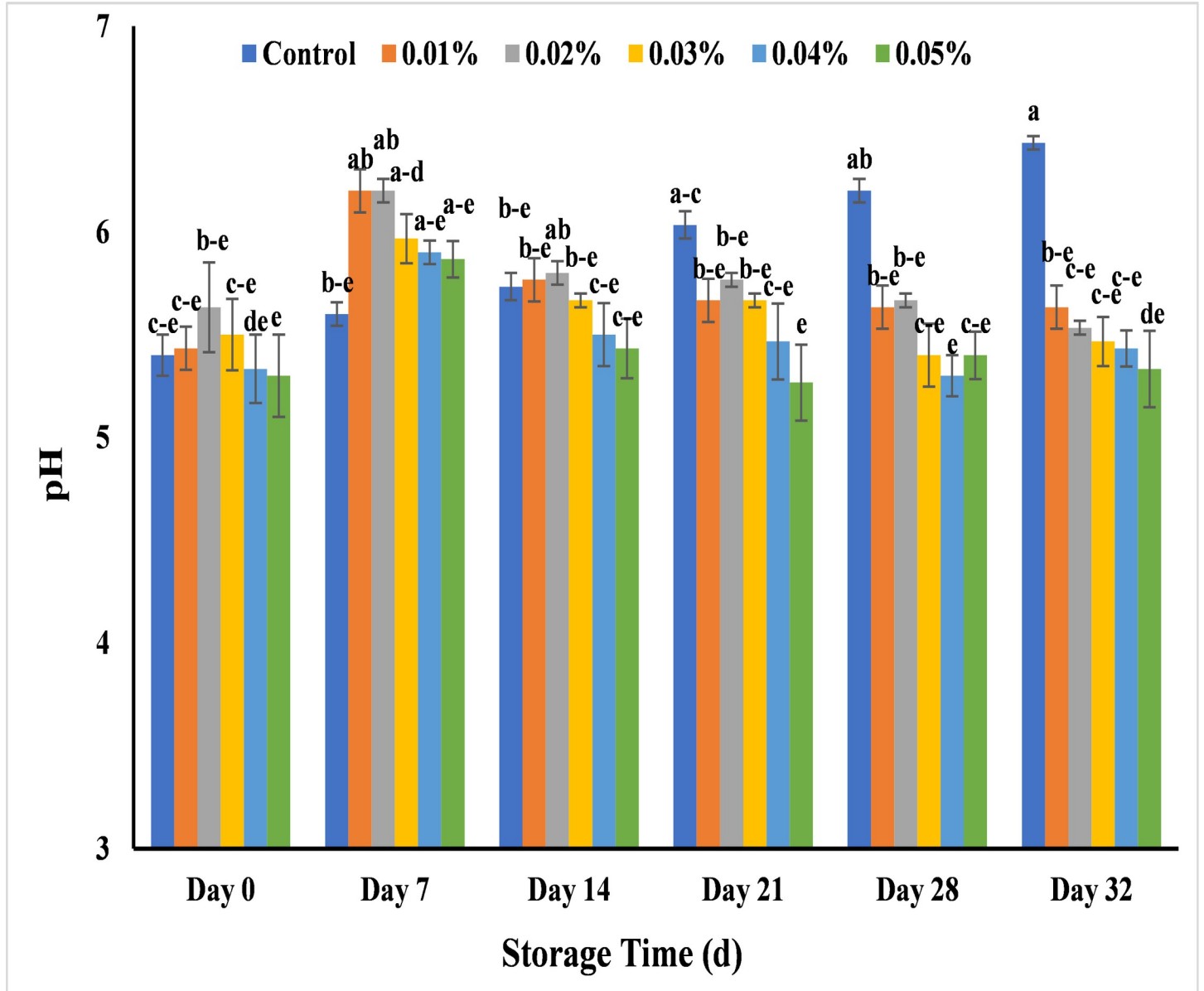

**Fig 16. Effect of AgNPs concentrations (T0 = control, T1 = 0.01%, T2 = 0.02%, T3 = 0.03%, T4 = 0.04%, T5 = 0.05%) on pH of Cavendish banana (BASRAI) during 32 days of storage.**

different climate conditions and type of fruit [70]. Generally, pH indicated the amount of organic acids in pulp of banana, which decreased at ripened stage of untreated banana due to use of these organic acids for respiration. However, bananas treated with AgNPs maintained the pH throughout the storage period. The acidity at un-ripen stage affects the taste of banana due to partial presence of oxalic acid which de-carboxylate at ripening stage by oxalate oxidase. So, control banana samples showed more change in pH as compared to treated banana samples. Similarly, different varieties of banana showed changes in pH depending on their ripeness and experimental conditions [33,69,70].

**Effect of AgNPs on titrable acidity (%) in banana.** Change in titrable acidity is linked to decay of banana. The change in titratable acidity of banana pulp was studied by applying

various concentration of AgNPs beside untreated bananas. A gradual increase in titrable acidity was detected in all treatments over the storage period of 21 days, where decline was noted after 21 days till 32 days of storage. Titrable acidity ranged from 0.32–0.33% in treated banana at day 0, that gradually increased with the increase of pH during 7 days of storage except untreated banana samples. It was observed that the untreated bananas showed decrease in titrable acidity making banana non-edible in 28 days of storage with maximum change in titrable acidity that was 0.16% while banana treated with AgNPs maintained the post-harvest quality of banana till 32 days storage period. It could be linked to the rise in malic acid, citric acid, and oxalic acid with start of ripening. However, main cause of the decline in acidity of banana at maturity was conversion of acid into sugar content [72]. Among all, lower change in titarable acidity was recorded with 0.01%, 0.02% and 0.03% AgNPs which was 0.33%, while gradual decrease in titrable acidity was recorded with 0.04% and 0.05% AgNPs that was 0.32% from day 0 to day 32 (Fig 17). Similarly in mango fruits coated with guar gum based AgNPs recorded less change in titrable acid values while untreated fruits had significantly more change at the end of the storage period [59]. In cherry tomatoes with Oolong tea-AgNps application showed stable acid contents at the end of 15 days storage as compared to chemically prepared AgNPs that showed only 3 days increase in shelf life [63]. The fluctuation in results was due to the application of silver nanoparticle coatings that is responsible to maintain acid contents in banana during storage [70]. Rapid increase in acid accumulation at immature stage raised the titrable acidity during ripening. Meanwhile the formation of sugar contents and physiological processes minimize the excessive increase of organic acids during maturity of banana [33,72,73].

**Effect of AgNPs on TSS (Brix) in banana (%).** TSS is the total amount of optically active compounds in fruits and vegetables, and it is an important indicator of fruit maturity. The flavor in fruit is mostly due to its titratable acids and total soluble solids (sugars). Sugars such as glucose, sucrose, and fructose are the major components of total soluble solids in banana pulp. Breakdown of starch into sugars during ripening accelerate the sugar content and decrease the starch content [38]. The analysis of total soluble solids (TSS) showed significant difference between treated and untreated banana. The results depicted that there was a high increase (12%) in TSS with untreated banana at day 14 and bananas were non-edible that reached to maximum at day 28 (19%) and bananas were fully decayed at 32 days of storage period followed by less increase in those bananas which were treated with different concentration of AgNPs. Among all, the lowest TSS was from 0.01%, 0.02% and 0.03% AgNPs that was 9.0%, 10.6% and 10.6% respectively while other 0.04% and 0.05% AgNPs showed a similar pattern of TSS as 11.3% and 13% (Fig 18). The more increase of TSS in untreated bananas was due to the formation of more organic solutes by moisture loss through water evaporation and conversion of these organic solutes into sugar contents [74]. The study was in line with the previous results according to that silver nanoparticle from grapefruit were used for the postharvest management of cucumber upto 21 days of storage [75]. Similarly, AgNPs were also used to increase the shelf life of banana for 5 days by maintaining the TSS in treated banana instead of control banana [22]. This increase in sugar content could also be attributed to conversion of starch into soluble sugars during ripening stage in the presence of ripening enzymes [35,76]. So, the silver nano-coating may reduce the fruit respiration rate and slow down the consumption of acid content during the physiological and metabolic processes of fruit. As a result, the AgNPs may prolong the ripeness of banana after harvesting and increase its shelf life [58,68].

**Effect of AgNPs on starch (%) in banana.** A quantitative study for starch analysis showed the maximum value during the early stage of storage in all banana samples in the range of 27–29. During 14 days, starch content showed maximum decrease with untreated banana and slight decrease with treated banana and on day 28 all controlled banana became rotten with

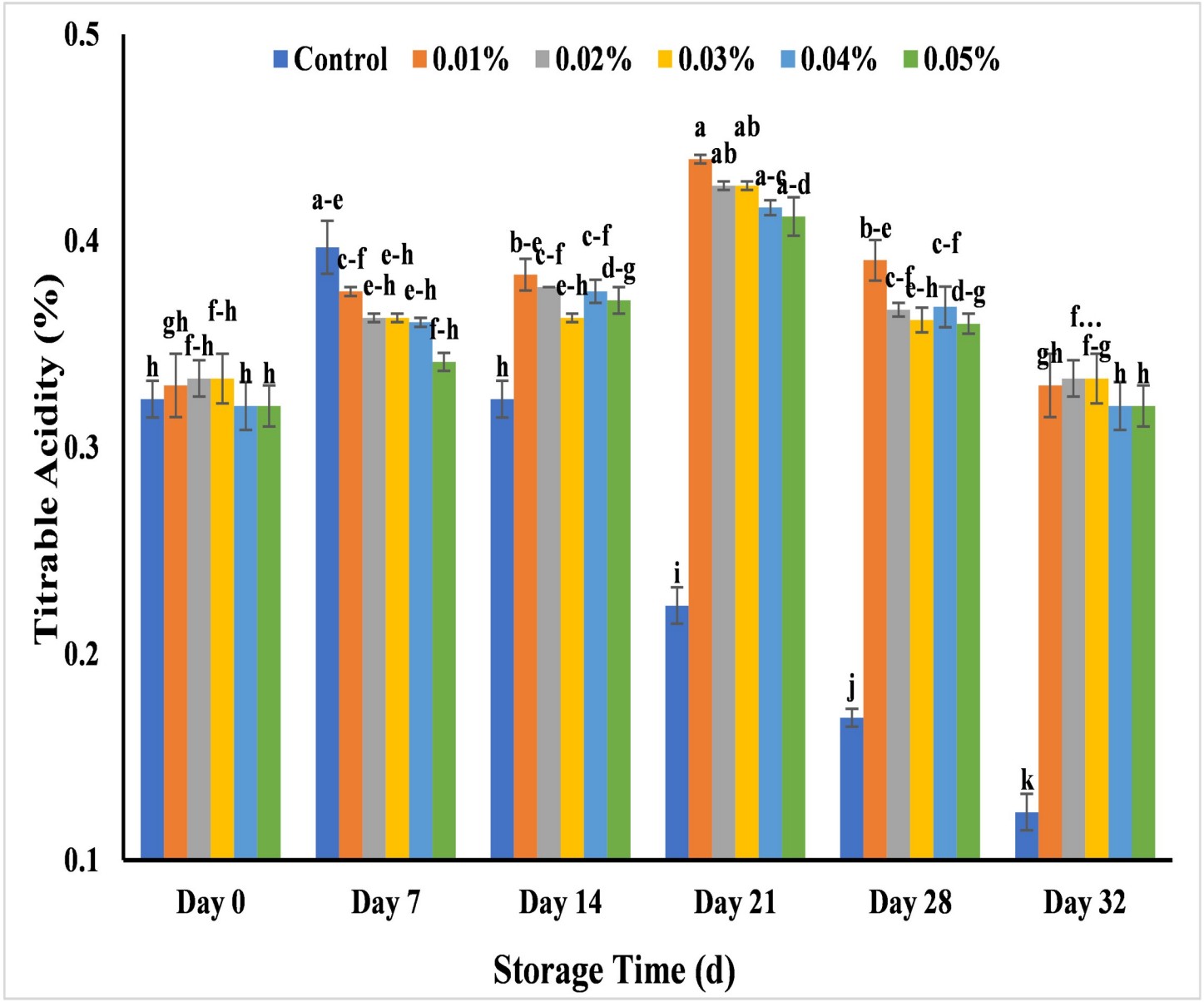

**Fig 17. Effect of AgNPs concentrations (T0 = control, T1 = 0.01%, T2 = 0.02%, T3 = 0.03%, T4 = 0.04%, T5 = 0.05%) on titrable acidity (%) of Cavendish banana (BASRAI) during 32 days of storage.**

starch content of 0.84%. Other than untreated bananas, all of the treated banana samples showed less change in the starch content till the end of storage period (day 32) with 0.01% AgNPs that was 4.44%. It was also noted that 0.02%, 0.03%, 0.04% and 0.05% showed more change in starch pattern as compared to 0.01% as the value was 3.45, 3.45, 3.44 and 3.44 respectively (Fig 19). Shelf life of tomato was increased by using AgNPs from tea extract. Maximum decline of starch was noticed in untreated samples rather than AgNPs treated sample that increased shelf life of tomato for 15 days in maturation period [63]. Lower starch values were noted in untreated banana samples due to the formation of soluble sugar by the hydrolysis of starch and high rate of respiration and moisture evaporation [69], while AgNPs control

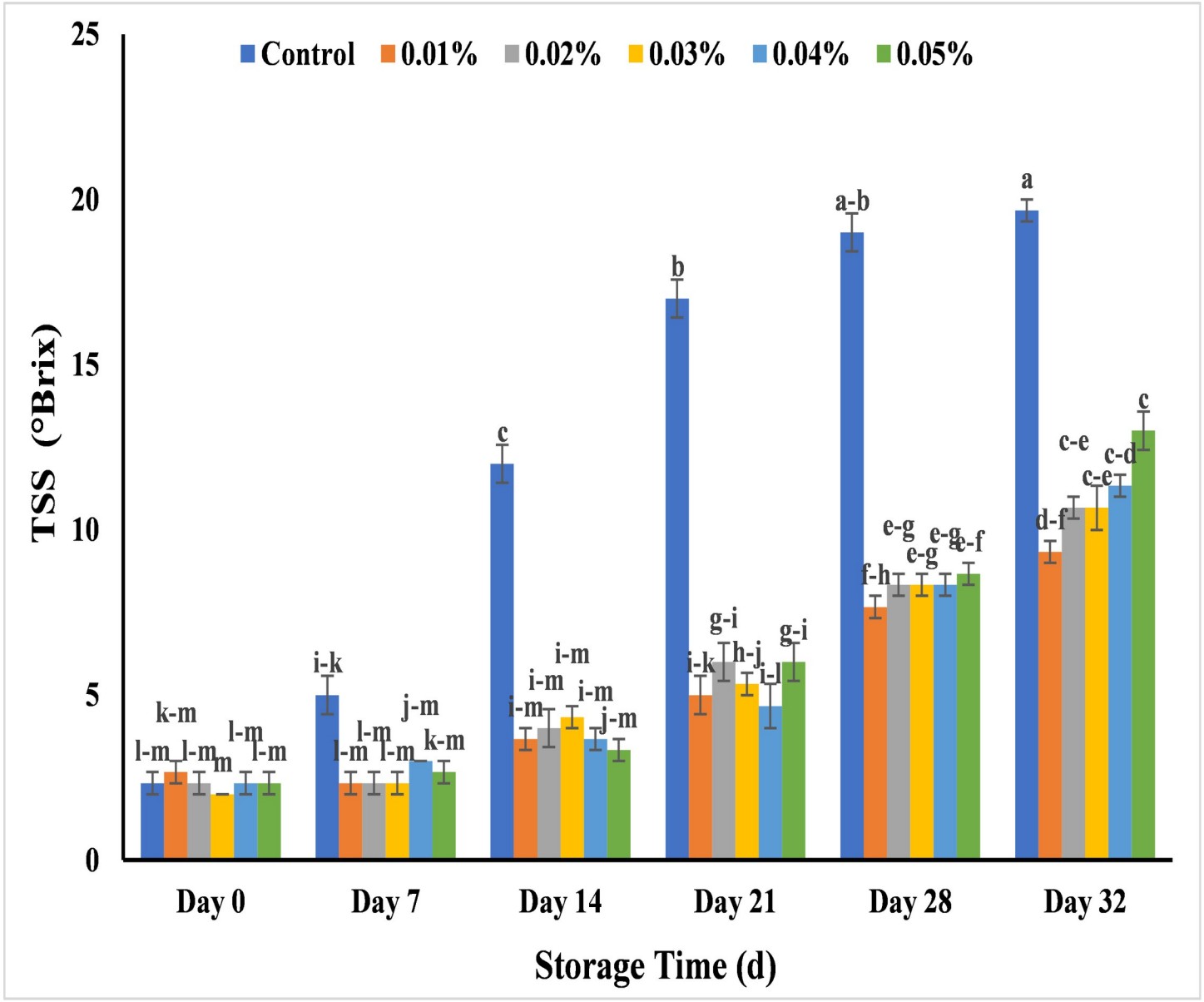

**Fig 18. Effect of AgNPs concentrations (T0 = control, T1 = 0.01%, T2 = 0.02%, T3 = 0.03%, T4 = 0.04%, T5 = 0.05%) on total soluble solids of Cavendish banana (BASRAI) during 32days of storage.**

respiration rate in peel by lowering oxygen level and increasing $CO_2$ level which is an inhibitor of starch breakdown [58,77,78].

**Effect of AgNPs on phenolic content (mg GAE/ 100 g) in banana.** Phenolic content involved in fruit shelf life act as stress defensive mechanism [79]. The study of phenolic content revealed a gradual increase in value from day 0 to day 21 and the slightly decrease till day 32 except un-coated banana. The banana without application of AgNPs started to become rotten showed a maximum decrease in phenolic content that was 33.3 mg GAE/ 100 g in day 21 and 26.7 mg GAE/ 100 g in day 32. All treated banana samples showed increase in phenolic content till the end of 21 days storage. In the start of banana ripening, the phenolic contents were recorded in the range of 26–29 mg GAE/ 100 g while at 21 days of storage, the phenolic

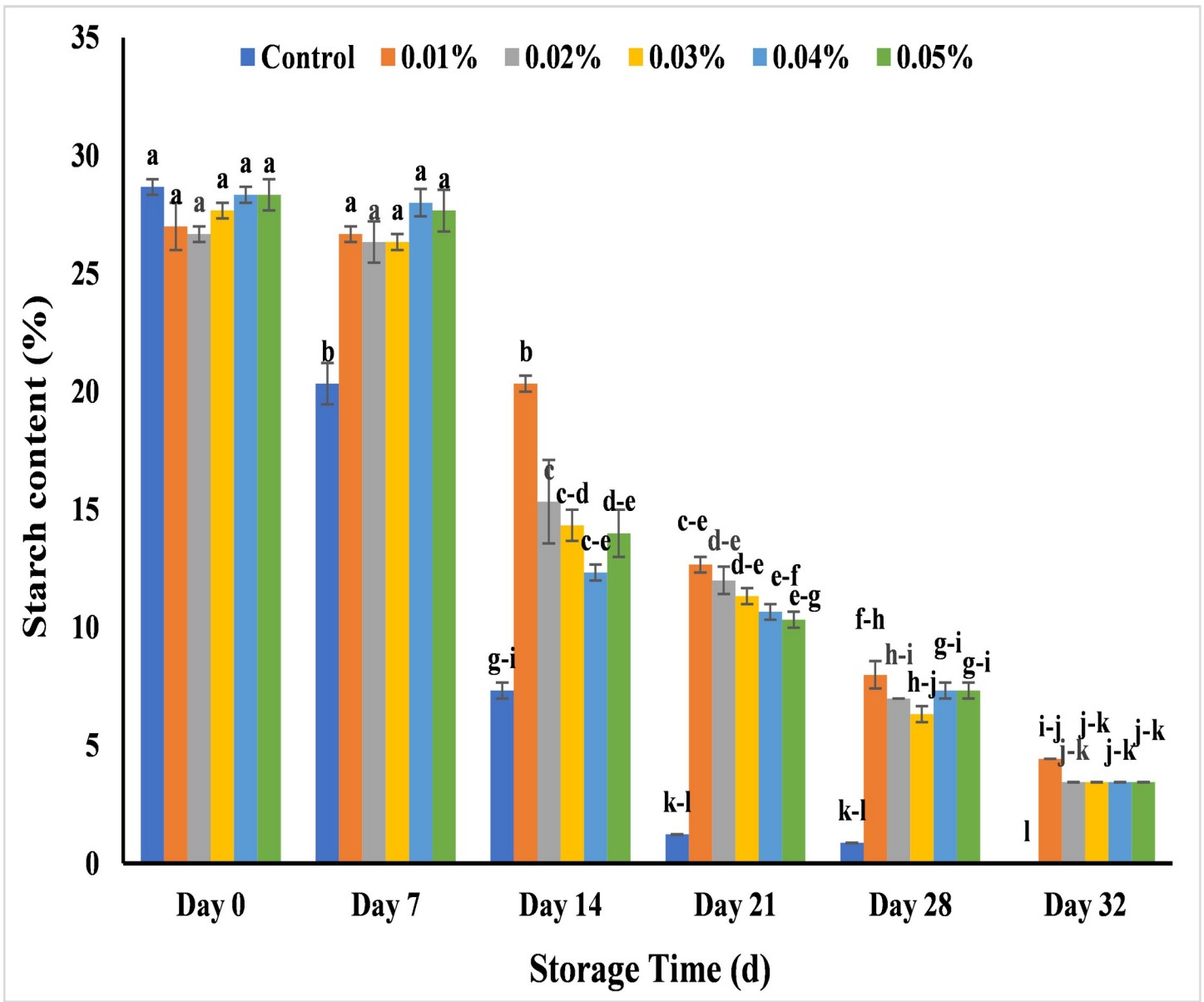

**Fig 19. Effect of AgNPs concentrations (T0 = control, T1 = 0.01%, T2 = 0.02%, T3 = 0.03%, T4 = 0.04%, T5 = 0.05%) on starch content of Cavendish banana (BASRAI) during 32days of storage.**

content were 0.01% (46.3 mg GAE/ 100 g), 0.02% (43.2 mg GAE/ 100 g), 0.03% (44.8 mg GAE/ 100 g), 0.04% (43.6 mg GAE/ 100 g), and 0.05% (43.5 mg GAE/ 100 g) while in end of storage, reduced decline was observed in phenolic contents with 0.01% (42.5 mg GAE/ 100 g), 0.02% (40.7 mg GAE/ 100 g), 0.03% (40.5 mg GAE/ 100 g), 0.04% (39.7 mg GAE/ 100 g), and 0.05% AgNPs (40.1 mg GAE/ 100 g) as compared to control sample (Fig 20). The previous literature supported the effect of AgNPs from longknog peel extract on the post-harvest quality of long-kong fruit and the significant lower decrease of phenolic content was noted with increase in storage for 9 days at 13˚C [73]. The lowest phenolic contents in untreated banana could be due to high respiration and oxidation mechanism that ultimately reduce the concentration of phenolic content that increase the risk of microbial attack and reduce the anti-oxidant ability in

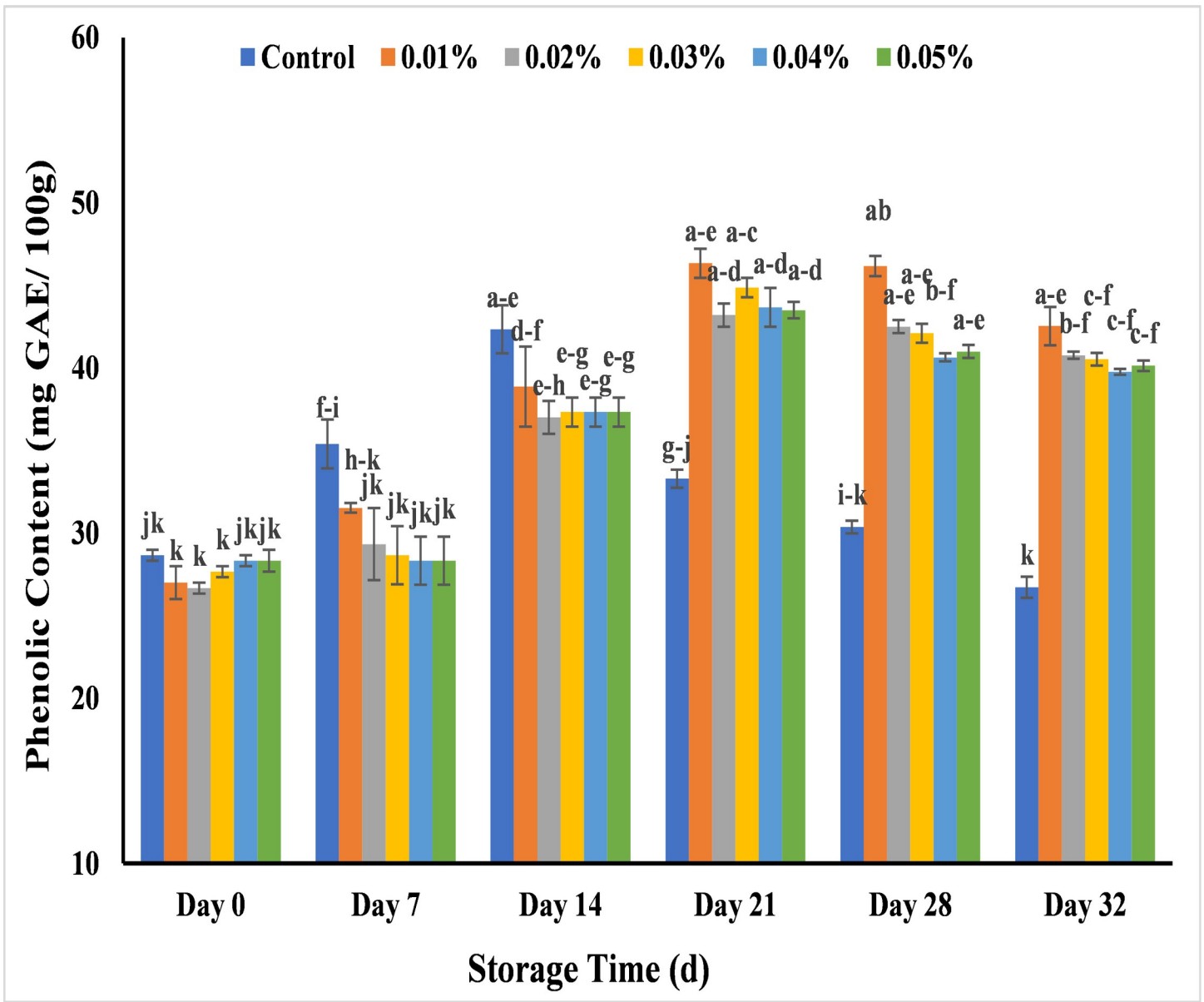

**Fig 20. Effect of various AgNPs concentration (T0 = control, T1 = 0.01%, T2 = 0.02%, T3 = 0.03%, T4 = 0.04%, T5 = 0.05%) on phenolic content of Cavendish banana (BASRAI) during 32 days of storage.**

banana by the activity of PPO enzymes. PAL is the key enzyme in biosynthesis of the phenolic compounds in the phenylpropanoid pathway. AgNPs induce the activity of PAL enzymes and showed increase initially [80] and then slight decline in phenolic content of treated banana as compared to control. Another enzyme PPO oxidized the phenolic content in untreated banana by using oxygen as co-substrate and the lower concentration of oxygen could be the reason to reduce phenolic content [70,73,81] while the AgNPs application on fruits enable them to maintain the phenolic content throughout the storage [38].

**Effect of AgNPs on protein (%) in banana.** The peak value in protein contents was noticed at the start of ripening (day 7) which in the range of 1.57–1.59%. The control banana sample attained the highest decrease in protein value on day 21 (1.36%), day 28 (1.23%) and

day 32 (0.95%) of storage and completely decayed during 32 days. It could be due to proteolysis in which protein breakdown and converted into amino acid by proteolytic enzymes for the use in metabolic process and sugar formation as the change in protein content indicates the nutritional maturity in banana and physiochemical changes [82]. As compared to control banana, other banana sample showed gradual decrease in protein content from day 0 to day 32 with 0.01%, 0.02%, 0.03%,0.04% and 0.05% AgNPs. The protein content value was 1.49% (0.01%), 1.41% (0.02%), 1.35% (0.03%), 1.33% (0.04%) and 1.33% (0.05%) at 32 days of storage. It was seen that banana samples showed minimum change in protein value by using 0.01% AgNPs as compared to other treatments (Fig 21). In previous research, the less change in protein contents under post-harvest conditions of rice over the 10 days treatment of AgNPs [37],

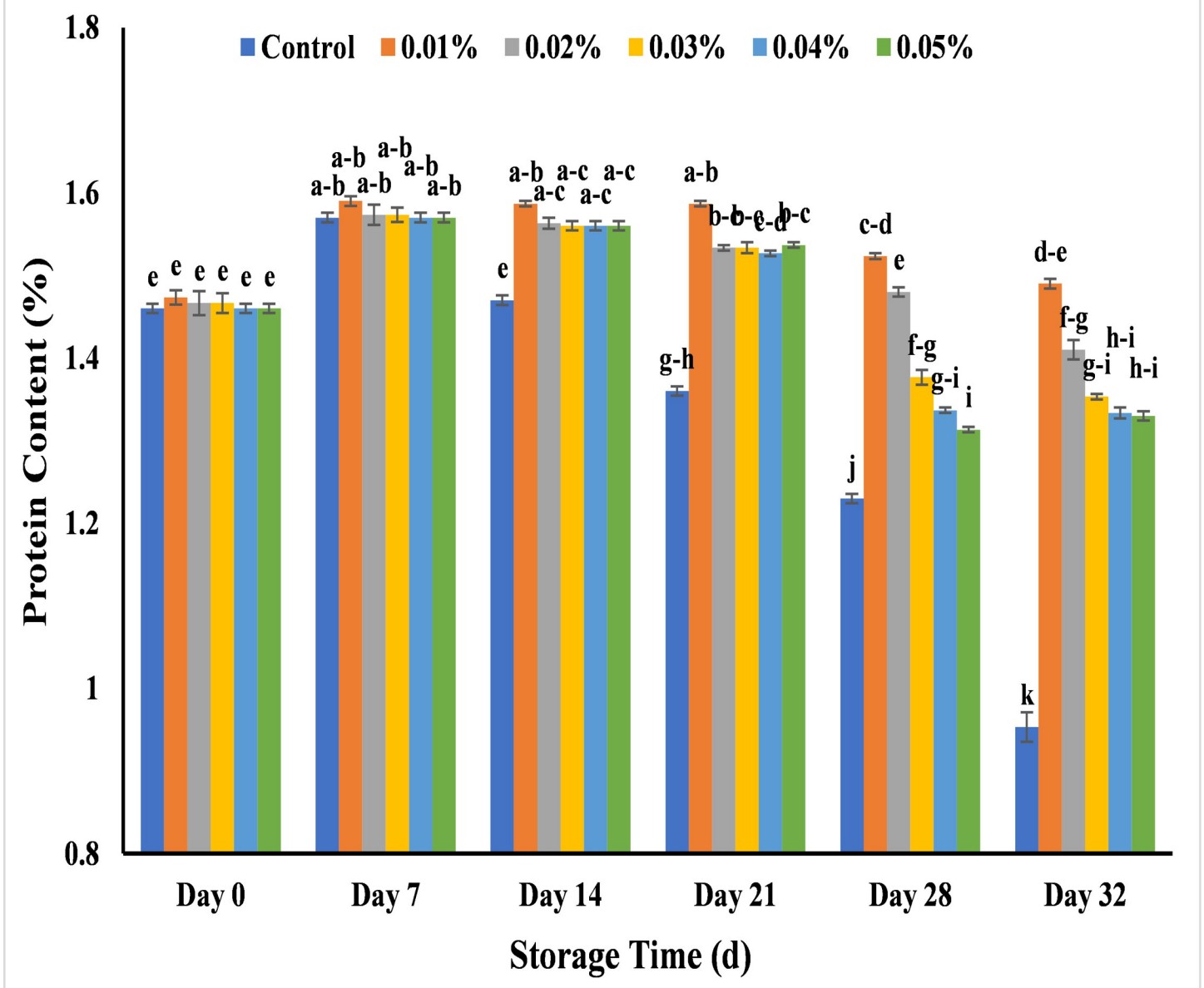

**Fig 21. Effect of various AgNPs concentration (T0 = control, T1 = 0.01%, T2 = 0.02%, T3 = 0.03%, T4 = 0.04%, T5 = 0.05%) on protein content of Cavendish banana (BASRAI) during 32 days of storage.**

cut flower of gladiolus over the 12 days treatment of AgNPS [83] and banana over the 30 days treatment of *Aloe vera* with AgNPs coatings [38] were studied by using AgNPs application as compared to control. The ability of AgNPs as semi permeable coating reduced respiration rate and change the internal atmosphere of banana by lowering the activity of those enzymes that are responsible for degradation of protein [37,83,84].

**Effect of AgNPs on ethylene production (ppm) in banana.** Ethylene plays an important role in the ripening of banana. In addition to increase the process of fruit ripening, ethylene frequently causes over-ripening and even rotting, which shortens shelf life of fruits and vegetables. Maximum increase in ethylene production was recorded during ripening of uncoated banana sample as it was 36.5 ppm on day 0 which reached 77.5 ppm on day 14 with maximum decay and 1129.2 ppm on day 32. This could be due to high respiration rate and autocatalytic ethylene production that cause the physiological and metabolic changes by change in chloroplast structure that reduce the chlorophyll content and increase decay in uncoated bananas [85]. Rather than control banana sample, treated banana depicted the less increase in ethylene rate from day 0 to day 32. All treated banana sample showed ethylene rate in rang of 35 to 38 ppm at day 0 while its production rate was 68.5 ppm with 0.01%, 70.3 ppm with 0.02%, 87.6 ppm with 0.03%, 89.4 ppm with 0.04% and 91.9 ppm with 0.05% AgNPs concentrations (Fig 22). Different concentrations of edible coating with chitosan nanoparticles suppressed the ethylene production and increase the shelf life of banana till 30 days of storage [38]. Similarly, another study investigated the use of guar gum based AgNPs to accelerate the shelf life of mango for 28 days at 25°C by reducing the ethylene production [59]. So, the AgNPs as fruit coating treatments act as a semipermeable membrane that reduce the respiration rate and ethylene production by altering internal atmosphere. It delay metabolic activity and potentially reduce the ripening process which ultimately lead to the increase in fruit storage life [59].

## Principle component analysis

Principal component analysis was done to provide better understanding of interaction within the results. The first two components showed a total variance of 91.7%. The first component, PC1 with a total variance of 78.8% mainly consisted of firmness, pH, starch, titrable acidity (TA), moisture and protein content on its negative axis and color, decay, weight loss percentage, total soluble solid (TSS), pulp to peel ratio, ethylene rate and phenolic content on its positive axis. The second component, PC2 with 12.9% of the variance made up the firmness, starch, moisture content, weight loss percentage, pulp to peel ratio and color were on positive axis while decay, pH, titrable acidity, total soluble solid (TSS), starch, phenolic and protein content were on negative axis (Fig 23). When the banana samples were accounted on the plane by the first two principal components PC1 and PC2, the different scoring positions were observed, depending on the treatments and storage week (Fig 24). It was noted that control banana samples appeared on the upper part of plane on positive axis of PC1 and PC2 with storage of 32 days which showed no increase in shelf life, while treated samples score appeared at the lower part of PC1 on negative axis from day 14 to 21 and on positive axis from day 28 to 32 which showed the increase in shelf life in order 0.01% > 0.02% > 0.03% > 0.04% > 0.05%. Silver NPs concentrations 0.04% and 0.05% showed score far away the positive axis of PC1 which represented the less increase in shelf life of banana as compared to other treatments. Among all treatments, 0.01% and 0.02% AgNPs showed the best scoring position that was near the positive axis of PC1 with less decay and maximum increase in shelf life. As the storage period progresses, the score of treated banana samples moves from negative axis to positive axis of PC1 and from positive axis to negative axis with PC2 as the shelf-life period showed a diagonal

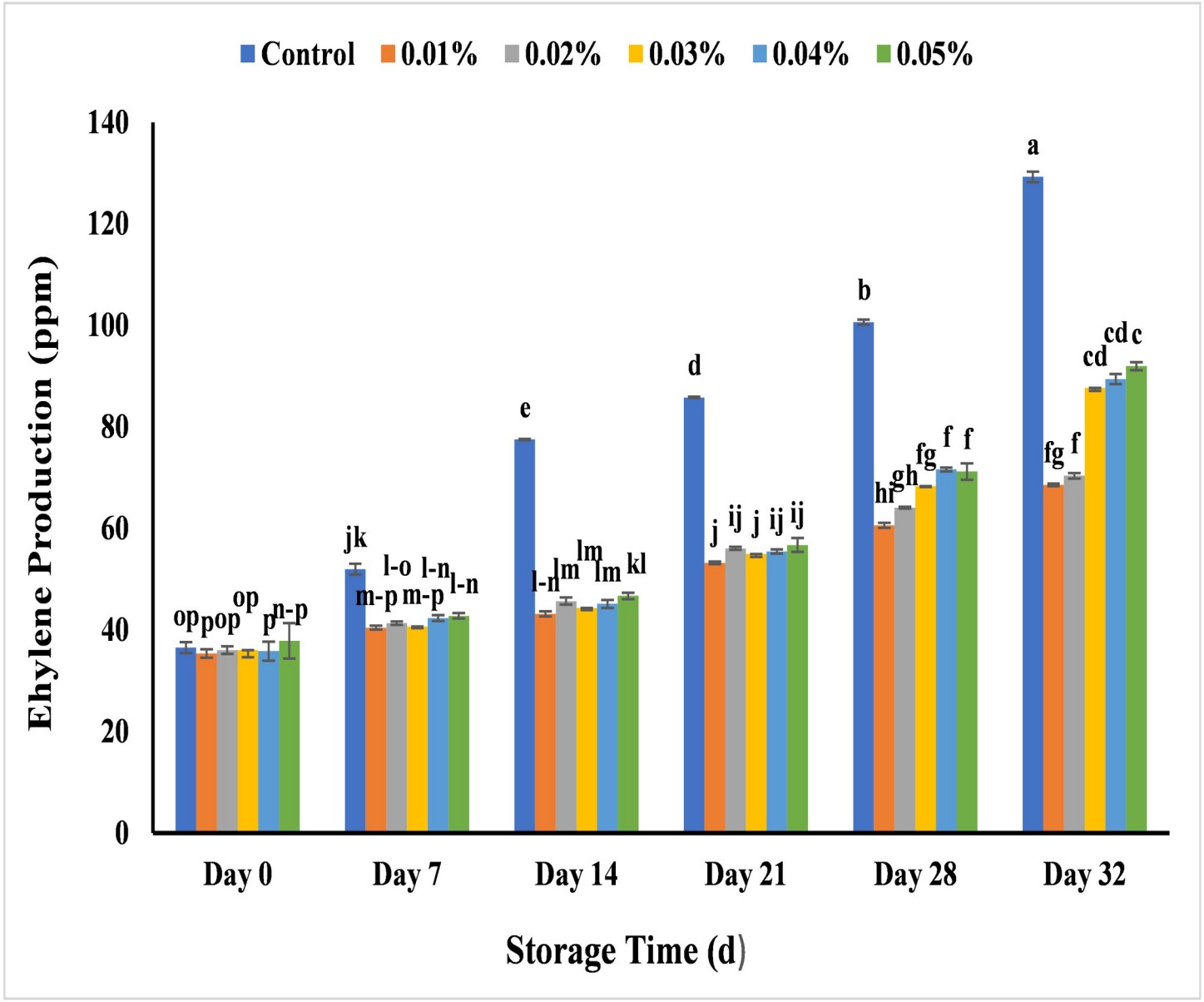

**Fig 22. Effect of various AgNPs concentration (T0 = control, T1 = 0.01%, T2 = 0.02%, T3 = 0.03%, T4 = 0.04%, T5 = 0.05%) on ethylene production in Cavendish banana (BASRAI) during 32 days of storage.**

upward shift in the PC1–PC2 plane. The similar results were observed with three cultivars of apricot to enhance its shelf life [59,86].

## Shelf life

Post-harvest losses is a major concern in food industry. Longer shelf life is recommended for the bestselling, storage, preservation, packaging and transportation of fresh fruit [87]. The results illustrate the impact of green synthesized AgNPs (0.01%, 0.02%, 0.03%, 0.04% and 0.05%) from ELE on the shelf life of banana stored on 32 days of storage. The maximum banana shelf life was attained till 32 days by using 0.01% and 0.02% AgNPs followed by 0.03% and 0.04% for 28 days, whereas the shortest shelf life was found at 14 days with untreated bananas

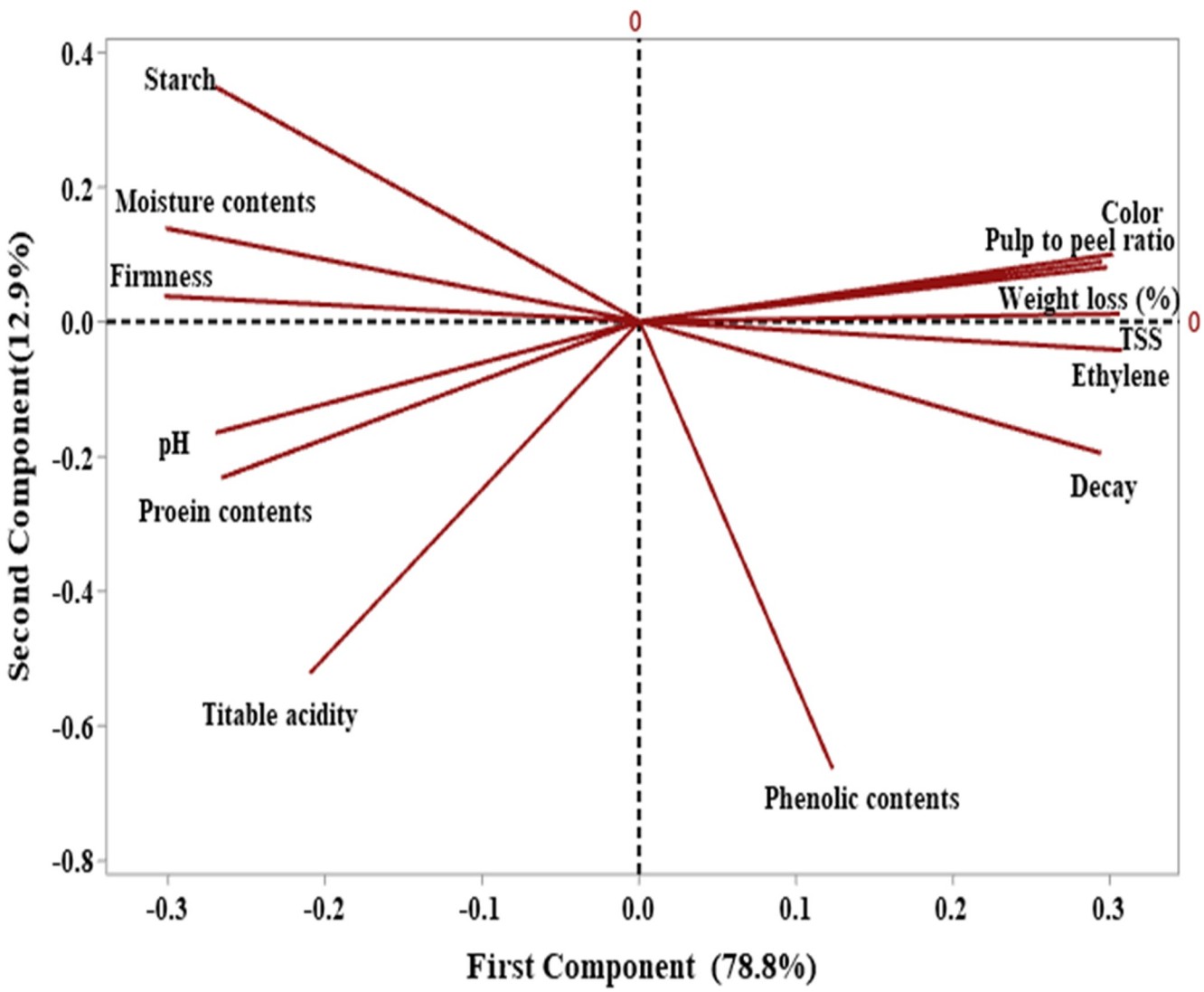

**Fig 23. Banana samples scores with different parameters in the PCA plane with PC1: 78.8% of total variance and PC2: 12.9% of total variance.**

(Fig 25). These nanoparticles serve as barrier to retard the morphological and physio-chemical ripening process by controlling the microbiological activity that may contribute to reduce the post-harvest losses of banana by increasing its shelf life. According to the previous results the use of chemically synthesized chitosan nanoparticles with Aloe vera and moringa coatings extended the shelf life of banana upto 30 days [38].Furthermore, it was also found that the silver NPs has superior effects as edible coatings throughout the storage period of mango upto 14 days [59] and strawberry upto 15 days [4] by extending their shelf life. So, this is the first study that report the 32 days shelf life of banana by using the green synthesized AgNPs as compared to previously reported increase in shelf life with chemical synthesized nanoparticles. Moreover, AgNPs has long term preservation (up to 1 year) and can be used for log time [88,89].

## Detection of AgNPs in banana pulp

Energy-dispersive X-ray spectroscopy (EDX) was used to study the silver element inside the banana pulp. Carbon, oxygen, magnesium, calcium, phosphorus, potassium and chlorine were

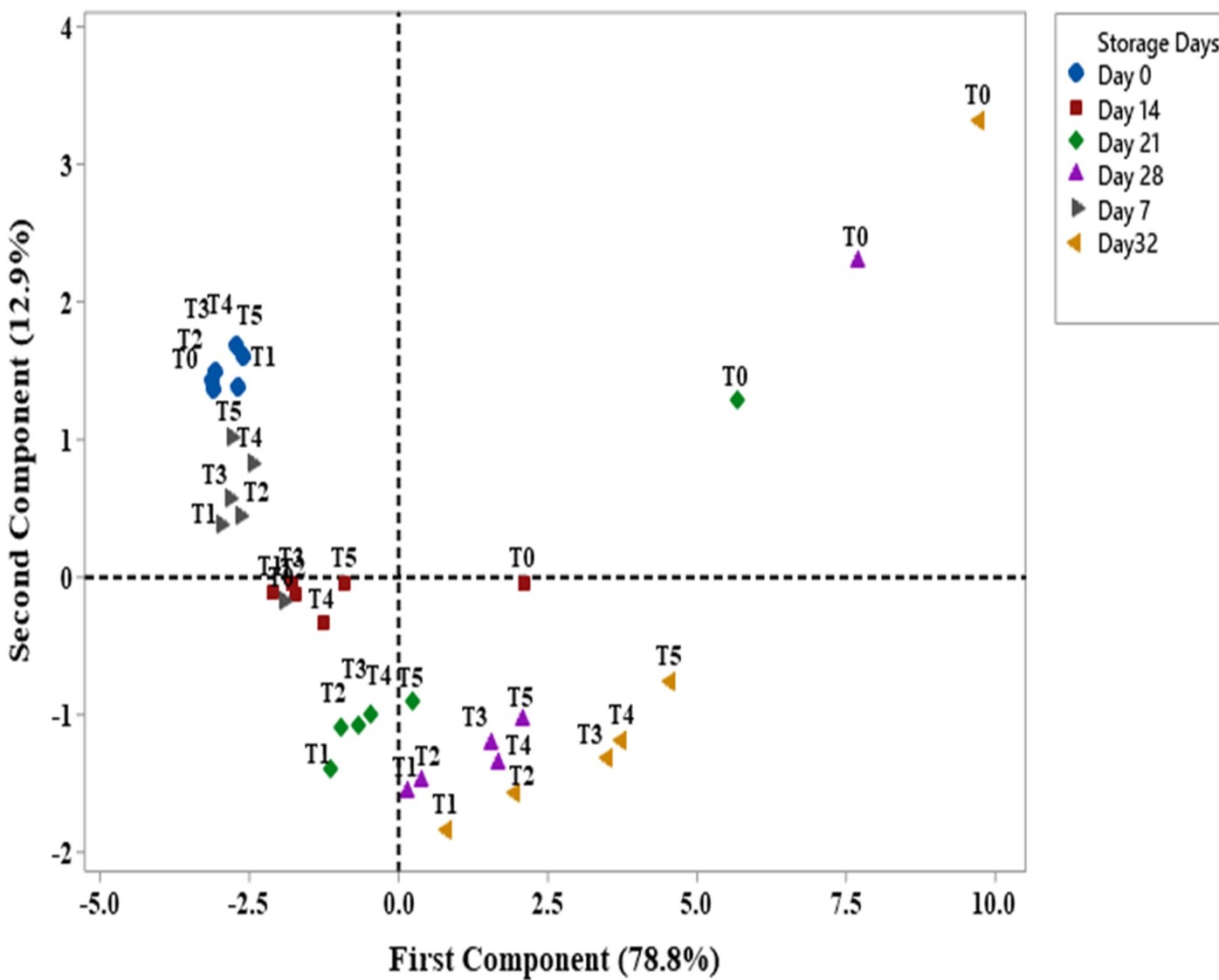

**Fig 24. Banana samples scores with various AgNPs concentration (T0 = control, T1 = 0.01%, T2 = 0.02%, T3 = 0.03%, T4 = 0.04%, T5 = 0.05%) in the PCA plane with PC1: 78.8% of total variance and PC2: 12.9% of total variance.** Symbols represent the storage days: 'Day 0' (●), Day 7 (►), Day 14 (■), Day 21 (♦), Day 28 (▲) and Day 32 (◄).

measured in samples at range of 0–4.0 keV (Fig 26). The analysis revealed that silver did not penetrate inside the banana pulp and banana treated with lower concentrations of AgNPs is safe to be used as food.

## Practical implications

The study aimed at addressing the cheap and long lasting methods to enhance the shelf life of Banana. Negative allelopathic effects of Eucalyptus have been reported. AgNPs synthesized from ELE can give dual benefits by enhancing shelf life of Banana and reducing allelopathic effects of Eucalyptus, as AgNPs can be stored upto 1 year or more without effecting their efficacy.

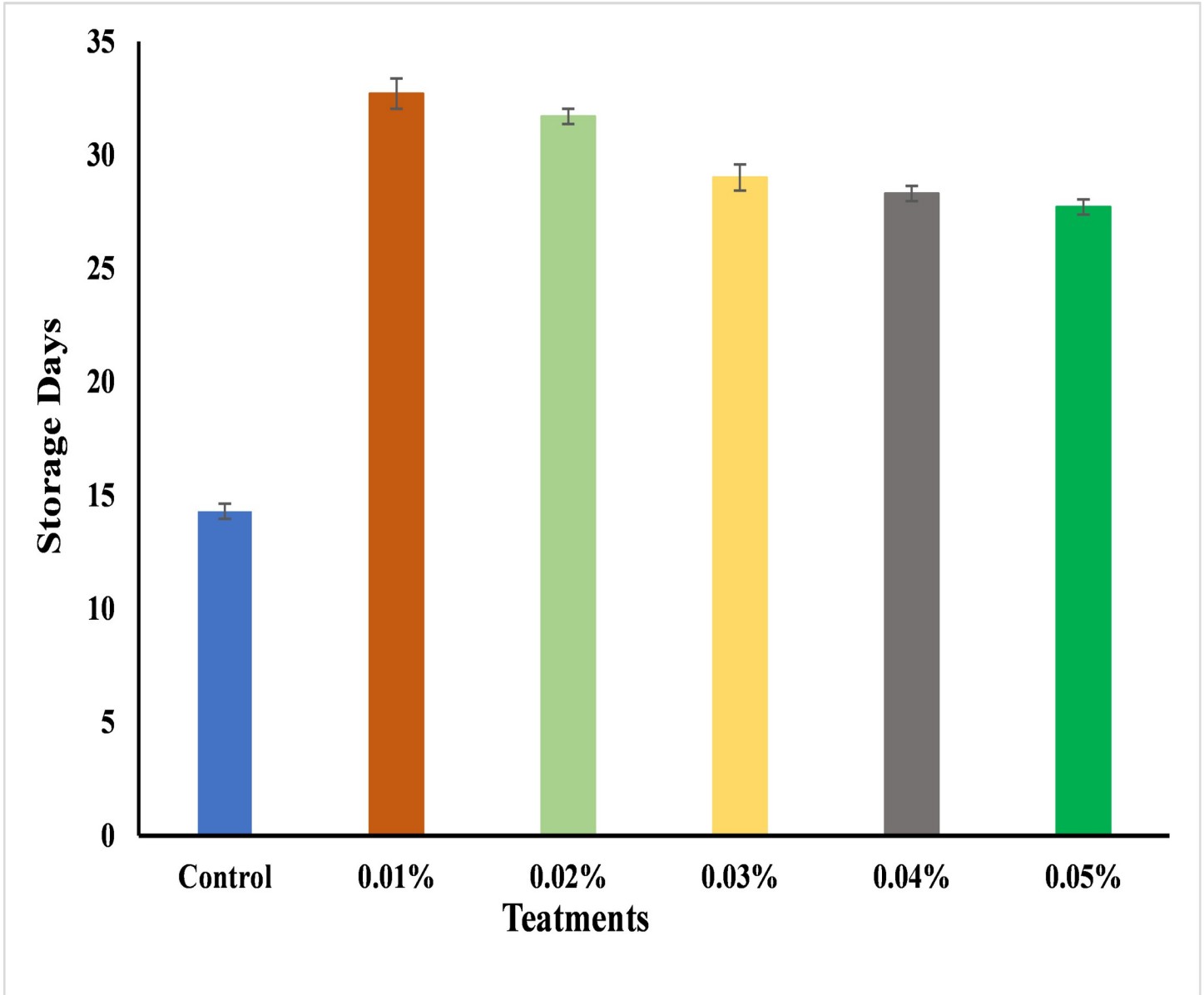

**Fig 25. The effect of AgNPs concentrations (T0 = control, T1 = 0.01%, T2 = 0.02%, T3 = 0.03%, T4 = 0.04%, T5 = 0.05%) on the shelf life of Cavendish bananas (BASRAI) stored at 25˚C from 0 to 32 days.**

## Conclusion

Shelf life of banana was increased upto 32 days at 25˚C by using 0.01% and 0.02% of AgNPs prepared from ELE. This is also supported by the less change in color, decay, firmness, pulp to peel ratio, weight loss, moistening contents, titrable acidity, pH, total soluble solids, ethylene production, phenolic and protein content of bananas during storage period as compared to untreated banana. Moreover, this study demonstrated that AgNPs enhanced the post-harvest quality of banana by exploring the green nanotechnology through leaves of eucalyptus tree to balance its ecological threat as it is unfit in future for arid and semi-arid regions. These findings indicated that the green synthesized nanoparticles as non-hazardous coating on banana

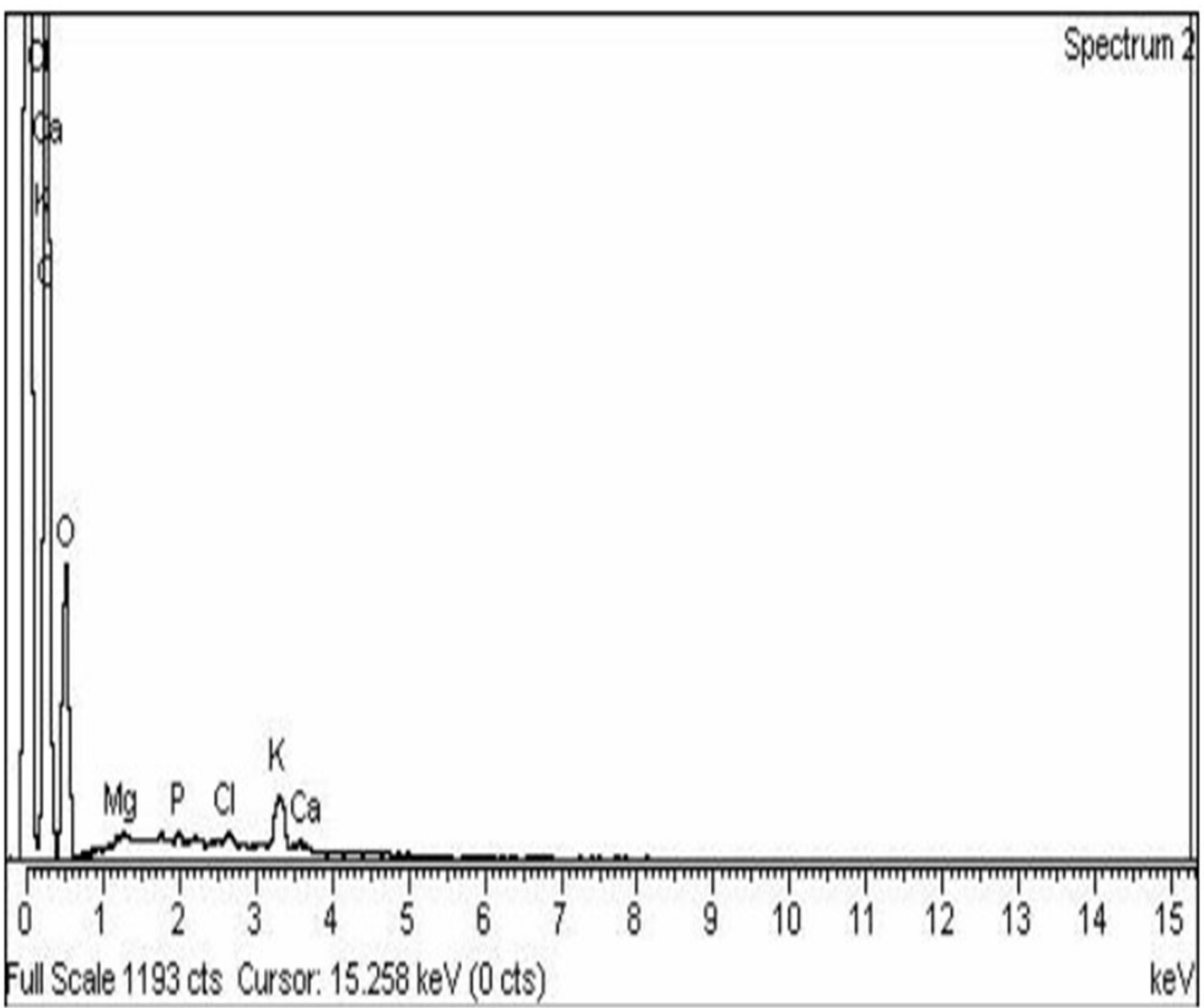

**Fig 26. EDX analysis to detect AgNPs penetration in banana pulp.**

that offer a great potential for postharvest management of agricultural products by preventing the distribution, supply and storage problem of tropical commodities. This approach will ultimately lead toward agro-industrial and eco-environmental sustainability and it will also reduce the economic burden of country. The approach is being novel as AgNPs can be stored for a longer period and Ag did not penetrated in the pulp, hence safe to be used as food.

## Author Contributions

**Conceptualization:** Shamim Akhtar.

**Data curation:** Durr-e- Nayab.

**Formal analysis:** Durr-e- Nayab.

**Methodology:** Durr-e- Nayab.

**Project administration:** Shamim Akhtar.

**Software:** Durr-e- Nayab.

**Supervision:** Shamim Akhtar.

**Writing – original draft:** Durr-e- Nayab.

**Writing – review & editing:** Shamim Akhtar.

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
