## [Decision Letter · Decision Letter 0]

7 Oct 2022

PONE-D-22-24409Green Synthesized Silver Nanoparticles from Eucalyptus Leaves can Enhance Shelf life of BananaPLOS ONE

Dear Dr. Akhtar,

Thank you for submitting your manuscript to PLOS ONE. After careful consideration, we feel that it has merit but does not fully meet PLOS ONE’s publication criteria as it currently stands. Therefore, we invite you to submit a revised version of the manuscript that addresses the points raised during the review process. Your manuscript is reviewed by two experts in the field. Both reviewers find the work interesting but raised multiple issues which need to be addressed properly in the revised version. The reviewers' reports can be found at the end of this email. 

We look forward to receiving your revised manuscript.

Kind regards,

Mohammad Irfan, Ph.D.

Academic Editor

PLOS ONE

Journal Requirements:

a. The name of the colleague or the details of the professional service that edited your manuscript.

b. A copy of your manuscript showing your changes by either highlighting them or using track changes (uploaded as a *supporting information* file).

c. A clean copy of the edited manuscript (uploaded as the new *manuscript* file).

"The funding for the research was provided by University of Gujrat"

"We acknowledge Department of Botany for funding the research, Dr. Muddasar Zafar, and Dr Tahir Iqbal

Awan for their timely support"

"The funding for the research was provided by University of Gujrat"

6. We note that you have indicated that data from this study are available upon request. PLOS only allows data to be available upon request if there are legal or ethical restrictions on sharing data publicly. For more information on unacceptable data access restrictions, please see http://journals.plos.org/plosone/s/data-availability#loc-unacceptable-data-access-restrictions. 

7. We note that you have stated that you will provide repository information for your data at acceptance. Should your manuscript be accepted for publication, we will hold it until you provide the relevant accession numbers or DOIs necessary to access your data. If you wish to make changes to your Data Availability statement, please describe these changes in your cover letter and we will update your Data Availability statement to reflect the information you provide.

8. Please include a separate caption for each figure in your manuscript.

Reviewers' comments:

Reviewer's Responses to Questions

**Comments to the Author**

1. Is the manuscript technically sound, and do the data support the conclusions?

Reviewer #1: Yes

Reviewer #2: Yes

2. Has the statistical analysis been performed appropriately and rigorously? 

Reviewer #1: Yes

Reviewer #2: I Don't Know

3. Have the authors made all data underlying the findings in their manuscript fully available?

Reviewer #1: Yes

Reviewer #2: Yes

4. Is the manuscript presented in an intelligible fashion and written in standard English?

Reviewer #1: Yes

Reviewer #2: Yes

5. Review Comments to the Author

Reviewer #1: I have various concerns before the manuscript can be reconsidered for publication in PLOS ONE. In the current article, the authors studied Manuscript PONE-D-22-24409, entitled "Green Synthesized Silver Nanoparticles from Eucalyptus Leaves can Enhance Shelf life of Banana"

Abstract: In abstract (P<0.05) should be written as P=0.05

What are the recommendations? Mention briefly in the abstract section.

Keywords: Keywords written should be different from the title section.

Introduction: In the first line of the introduction Musa spp. should be italicised. Why the deterioration of bananas during post-harvest takes place…should be mentioned in the introduction section. The introduction needs to be revised so that the hypothesis is explicitly clear.

Material and methods:

Check the space between numerical values and units to correct it throughout the manuscript.

Use a similar pattern of units throughout the manuscript.

On page 5, I suggest to expand the silver nanoparticle at once and then using an abbreviation of AgNPs throughout the manuscript.

In section 2.6, the morphological studies section mention units in the subtitle of each parameter.

Correct the serial number of subtitles after 2.6.1.5 Pulp to peel ratio.

In 2.6.1.7. Titrable acidity: rewrite the sentence. For titration NaOH (0.1N solution was used until it turned light pink colour. Not clear.

On page 6 Treatments are not properly mentioned. Kindly mention the treatments in a proper way as these are not clear. Mention it as T0 (Control), T1, T2 so on….

Correct the subtitle effect on phenolic content of banana to the measurement of the phenolic content of banana.

On pages 7 and 8 starch content, total soluble sugar and protein estimation are represented in which units kindly make clear in the material and method section.

Write content instead of contents.

Results and discussion section:

On page 12 section 3.7.5 authors have mentioned the units of ratio…according to me the ratio is unitless kindly check and correct it.

As result in some places, authors have written day 31 or day 32. Make it clear throughout the manuscript.

On page 14 section 3.7.8 first line corrects titratable to Titrable.

Words having abbreviations must be expanded at once and after that, their abbreviation must be followed. Correct it throughout the manuscript.

The discussion is vague. There is a lack of connection between cause-and-effect responses. It should be improved by linking the data and crafting the writing.

Conclusion: In conclusion, the authors mentioned five varieties of bananas but throughout the manuscript, I did not see any variety discussion.

References: References must be carefully checked and the scientific name in each reference should be italicised

Figures: There are so many figures which can be merged into subsections or can be added as supplementary files. Figures 6-10 should be shown as time-based bar graphs with treatments to be able to see the effects of treatments better, not clear as they are now. For clear statistical differences a, b, and c should be added

Reviewer #2: The manuscript entitled “Green Synthesized Silver Nanoparticles from Eucalyptus Leaves can Enhance Shelf life of Banana” explained the effects of the sliver nanoparticles extract on the quality, fruit composition, and shelf life of banana fruit. The authors found that all tested concentrations form the silver nanoparticles maintained fruit quality and shelf life compared with untreated fruit. The authors showed that the treatments did not accumulate any residues in the fruit flesh and they suggested that the treatment might be safe for human consumption.

The study is interested and can contribute to enrich the literature of banana storage. However, there are some comments and suggestions should be addressed to improve the manuscript.

1- The first sentence in the materials and method “Present study comprehends the green synthesis of nanoparticles from eucalyptus leaves to check their effect on the post-harvest losses in banana and its management” Move it to objective paragraph.

2- In the sample collection part: add the age of the used trees for more clarity.

3- Add the manufacturer and the country for the spectrophotometer.

4- In the part of measuring starch content: what do you mean with hot ethanol? Ethanol evaporates at 60°C. Please more clarification is needed.

5- In the statistical analysis part: add the country of the used software and explain about the PCA.

6- Since you know from beginning that the silver might inhibit the ethylene production in banana, why you did not measure ethylene from fruit? More explanation is needed.

7- Add the significant effect between treatments to the text and as letters to the figure in all parts for more clarity

8- In the part of “3.7.3. Effect of AgNPs on Banana Firmness (N)” correct the typo mistake for “understand” in line 9.

9- It was unclear how the control fruit had the highest weight loss and the highest moisture content in the same time compared with other treatments! Please, more clarification is needed.

10- In part of “7.7. Effect of AgNPs on pH of Banana” I suggest to delete the first sentence.

11- It was not clear to me how TA increased during shelf life and usually the TA decreased by increasing the fruit maturity and senescent. Please more clarification is needed.

12- add the “doi” to all references.

6. PLOS authors have the option to publish the peer review history of their article (what does this mean?). If published, this will include your full peer review and any attached files.

Reviewer #1: **Yes: **Bhavya Bhargava

Reviewer #2: No

---

## [Author Response · Author response to Decision Letter 0]

20 Dec 2022

S.NO. Reviewer Comment Response Page no/Line No

1. Reviewer 1 Abstract: In abstract (P<0.05) should be written as P=0.05 Statistically significant results were recorded (P=0.05) by applying five different concentrations of silver nanoparticles (AgNPs) in ranges of 0.01-0.05%. Page 1 and line 15-16

2. Reviewer 1 What are the recommendations? Mention briefly in the abstract section. It is recommended to use 0.01% AgNPs to enhance the shelf life of banana without effecting its nutritive value.

 Page 1 and line 23-24

3. Reviewer 1 Keywords: Keywords written should be different from the title section. Post-harvest decay, Basrai, Ripeness of banana, Economic losses, Ecological threat, Ethylene Page 1 and line 25

4.

 Reviewer 1 Introduction: In the first line of the introduction Musa spp. should be italicized Banana (Musa spp.) is an important commercial fruit that has a great economic value due to its high consumption demand. Page 1 and line 28

5. Reviewer 1 Why the deterioration of bananas during post-harvest takes place…should be mentioned in the introduction section. The introduction needs to be revised so that the hypothesis is explicitly clear. Bananas are transported from their production areas to distant locations for marketing and consumption. If post-harvest bananas are not treated appropriately, they become susceptible to damage and degradation during transportation, marketing and storage. Bananas are physiologically sensitive to decay after harvest due to continuous change in metabolic processes such as transpiration or respiration rate (3). Physical injuries and enzymatic actions by microorganism attack, or a combination of these factors can cause damage and degradation. Banana injuries and damage may result in moisture loss due to more surface evaporation (4) as well as microorganisms (fungi, bacteria) attack on injured fruits, causing substantially faster respiration rate than that of healthy bananas. Fruits with faster respiration and metabolic activity result in early storage decay or rots (5). However, it suffers from many post-harvest nutritional losses which cause physiological and morphological changes such as color change, decay, weight loss, loss of starch, protein and phenolic contents (6,7). For this reason, number of technologies are in practice to extend the shelf life of bananas by subjecting the controlled environmental conditions as low temperature storage, modify the atmospheric condition of storage and packing but these techniques are highly expensive Page 1-2 and line 31-44

6. Reviewer 1 Material and methods:

Check the space between numerical values and units to correct it throughout the manuscript.

Use a similar pattern of units throughout the manuscript. The manuscript was checked and the issue has been resolved Throughout the manuscript

7. Reviewer 1 On page 5, I suggest to expand the silver nanoparticle at once and then using an abbreviation of AgNPs throughout the manuscript. The manuscript was checked and the issue has been resolved Throughout the manuscript

8. Reviewer 1 In section 2.6, the morphological studies section mention units in the subtitle of each parameter. The manuscript was checked and the issue has been resolved Throughout the manuscript

9. Reviewer 1 Correct the serial number of subtitles after 2.6.1.5 Pulp to peel ratio. The sectioning has been removed according to the format Page 6 and line 164

10. Reviewer 1 In 2.6.1.7. Titrable acidity: rewrite the sentence. For titration NaOH (0.1N solution was used until it turned light pink colour. Not clear. Titrable acidity of banana pulp was measured by direct titration method in which 10 mL pulp was taken in beaker and 2-3 drops of phenolphthalein (1% solution) were added to it. For titration, NaOH (0.1 N solution in burette) was added dropwise in beaker until the color of sample turned light pink. Titrable acidity was calculated by using the following formula (33).

 Page 6 and line 178-181

11. Reviewer 1 On page 6 Treatments are not properly mentioned. Kindly mention the treatments in a proper way as these are not clear. Mention it as T0 (Control), T1, T2 so on….

 Cavendish bananas (Basrai) were collected from banana field Tando Jam, Sindh and each set had six bananas per treatment beside an uncoated (T0=control) set. Banana is widely grown for research purpose by Pakistan Agriculture Research Council, Islamabad, and were available on request without any permission. The banana was dipped in different concentration of AgNps solution (T1=0.01%, T2=0.02%, T3=0.03%, T4=0.04%, T5=0.05%) for 5 minutes. Page 5 and line 145-148

12. Reviewer 1 Correct the subtitle effect on phenolic content of banana to the measurement of the phenolic content of banana. Measurement of phenolic content (mg GAE/ 100g) of banana

 Page 7 and line 203

13. Reviewer 1 On pages 7 and 8 starch content, total soluble sugar and protein estimation are represented in which units kindly make clear in the material and method section. 

Write content instead of contents. Measurement the starch content (%)

To study the starch conversion, anthrone test was conducted. For this reason, the 0.5g banana pulp was homogenized with 5ml 80% ethanol in air tight beaker and incubated in water bath for 30 minutes at 80°C. Then it was centrifuged at 44000 rpm for 5 minutes. After that, 20 mL of distilled water and 6.5mL of perchloric acid was added in it. Then centrifuged at 4°C for 20 minutes and the supernatant was saved as extract. This process was repeated second time by adding 5mL distilled water and the supernatant was added in first extract by making the volume upto 100 mL with distilled water. Then 0.1 mL was taken and final volume was made upto 1mL by adding distilled water. Absorbance was measured at 620 nm by single beam spectrophotometer. Further, stock solution of glucose was prepared by adding 100mg glucose in 100 mL distilled water and standard solution was prepared by diluting the 10 mL of stock solution by making final volume upto 100 mL by using distilled water. The standard curve was drawn by using different concentration of standard solutions (0.1, 0.4, 0.6, 0.8, 1 mL) in each test tube and by making the volume of 1 mL. Then, 4mL of anthrone reagent was added to each test tube and heated them for 8 minutes. After cooling, 5mL distilled water was mixed with it. To measure the starch content, the value found for glucose was multiplied by 0.9 (34).

Total soluble sugar content (%)

To access TSS, 15 g banana pulp from each treatment was blended with 45 mL distilled water and few drops of extract were placed on refractometer prism after centrifugation at 11000 rpm for 5 minutes. Brix reading (sugar contents in aqueous medium) was calculated by calibration of refractometer with distilled water (35).

Measurement of phenolic content (mg GAE/ 100g) of banana

Folin Ciocalteu (F.C.) colorimetric method was used to detect the phenolic content in banana. For this purpose, 0.5 mL banana extract was added in 0.5 mL F.C. reagent and it was homogenized manually for 20s. After 3 minutes, 7% sodium carbonate solution of 2 mL was added in control and experimental test tubes and were placed in boiling water for 1 minute. After cooling the mixture, absorbance was measured by single beam spectrophotometer at 760 nm. The results were presented in mg of Gallic acid equivalents (mg/10 g). A calibration curve was plotted by using different concentrations of standard solution of Gallic acid (20-100 mg/L) (36). 

Protein estimation (%)

Protein content in bananas was estimated by adding 0.25 g pulp extract in 10mL of potassium phosphate buffer (50 mM) by adjusting its pH 7.8 at room temperature. After that, all samples were centrifuged for 15 minutes and 2mL Bradford reagent was added in 0.1 mL aliquot of each sample.

 Absorbance of each sample was measured by spectrophotometer at 590 nm (37).

 Page 7-8 and lines 184-215

14. Reviewer 1 Results and discussion section:

On page 12 section 3.7.5 authors have mentioned the units of ratio…according to me the ratio is unitless kindly check and correct it. Effect of AgNPs on pulp to peel ratio (%) of banana

Pulp to peel ratio is a consistent index to study the post-harvest losses during ripening of banana, that also reveal the change in their moisture contents (64). The effect of various AgNPs coatings on pulp to peel ratio of bananas showed the range of 1.2-1.3% with all treatments at the start of experiment and the treated banana with 0.01%, 0.02% and 0.03% had more effect on percentage of pulp to peel ratio as compared to 0.04% and 0.05%, Basrai showed less increase in pulp to peel ratio percentage that was 1.4-1.6% at day 7 and 1.9-2.4% at day 32 . As clearly observed in (Fig 7A) the pulp to peel ratio of uncoated banana increased rapidly that was 2.4% on day 14 and 3.4% on day 32 with control. So, it was noted that when the peel weight was divided by the pulp weight in the current study, the pulp to peel ratio of bananas showed more upward tendency in untreated banana as compared to treated bananas. This might be due to water loss from peel to pulp and atmosphere as well as the increased amount of soluble sugar in pulp ultimately increased the pulp to peel ratio by transferring the osmotic pressure in the pulp more quickly (65) . Similarly, the pulp to peel ratio was 1.6% with 0.01%, 0.02%, 0.03%, 0.04% and 1.7% with 0.05% AgNPs on day 14. It was revealed that all banana samples with AgNPs treatments got less increase in pulp to peel ratio at 32 days of storage in which 0.03 % (2.1%), 0.04% (2.1%) and 0.05% (2.5%) that was less adequate for post-harvest quality of banana as compared to 0.01% (1.8%) and 0.02% (1.9%) with more appropriate results (Fig 7A). The results are similar to the work in which 1.15% and 1.25% concentration of chitosan NPs increased the shelf life of banana upto 11 days with higher pulp to peel ratio in un treated banana as compared to treated banana samples (8).

Page 13 and line 381-399

15. Reviewer 1 As result in some places, authors have written day 31 or day 32. Make it clear throughout the manuscript.

 Basrai showed less increase in pulp to peel ratio percentage that was 1.4-1.6% at day 7 and 1.9-2.4% at day 32.

Whole manuscript has been checked Page 13and line 386-387

16. Reviewer 1 On page 14 section 3.7.8 first line corrects titratable to Titrable.

 Change in titrable acidity is linked to decay of banana. The change in titratable acidity of banana pulp was studied by applying various concentration of AgNPs beside untreated bananas. A gradual increase in titrable acidity was detected in all treatments over the storage period of 21 days, where decline was noted after 21 days till 32 days of storage. Page 15 and line 445-448

17. Reviewer 1 Words having abbreviations must be expanded at once and after that, their abbreviation must be followed. Correct it throughout the manuscript. The manuscript was checked and the issue has been resolved Throughout the manuscript

18. Reviewer 1 The discussion is vague. There is a lack of connection between cause-and-effect responses. It should be improved by linking the data and crafting the writing. The manuscript was checked and the issue has been resolved In the Result and discussion section

19. Reviewer 1 Conclusion: In conclusion, the authors mentioned five varieties of bananas but throughout the manuscript, I did not see any variety discussion.

 Conclusion

Shelf life of banana was increased upto 32 days at 25oC by using 0.01% and 0.02% of AgNPs prepared from ELE. This is also supported by the less change in color, decay, firmness, pulp to peel ratio, weight loss, moistening contents, titrable acidity, pH, total soluble solids, ethylene production, phenolic and protein content of bananas during storage period as compared to untreated banana. Moreover, this study demonstrated that AgNPs enhanced the post-harvest quality of banana by exploring the green nanotechnology through leaves of eucalyptus tree to balance its ecological threat as it is unfit in future for arid and semi-arid regions. These findings indicated that the green synthesized nanoparticles as non-hazardous coating on banana that offer a great potential for postharvest management of agricultural products by preventing the distribution, supply and storage problem of tropical commodities. This approach will ultimately lead toward agro-industrial and eco-environmental sustainability and it will also reduce the economic burden of country. The approach is being novel as AgNPs can be stored for a longer period and Ag did not penetrated in the pulp, hence safe to be used as food.

 Page 20 in the conclusion section

20. Reviewer 1 References: References must be carefully checked and the scientific name in each reference should be italicized

 The reference section was checked and the issue has been resolved In reference section

21. Reviewer 1 Figures: There are so many figures which can be merged into subsections or can be added as supplementary files. All figures have been added in separate figure file according to format and only captions are added in manuscript In Result and discussion section 

22. Reviewer 1 Figures 6-10 should be shown as time-based bar graphs with treatments to be able to see the effects of treatments better, not clear as they are now. For clear statistical differences a, b, and c should be added All figures were changed into time-based bar graphs with treatments and statistical differences a, b and c has been added. In separate figure file as per format

23. Reviewer 2 1- The first sentence in the materials and method “Present study comprehends the green synthesis of nanoparticles from eucalyptus leaves to check their effect on the post-harvest losses in banana and its management” Move it to objective paragraph. Present study comprehends the green synthesis of nanoparticles from eucalyptus leaves to manage post-harvest losses in banana. The aim was extended to optimize different factors as silver nitrate and ELE to synthesize green route AgNPs for its better assembly and characterization by using various analytical techniques as UV-vis spectroscopy, X-ray diffraction (XRD), scanning electron microscopy (SEM), transmission electron microscopy (TEM) and Fourier transform infrared spectroscopy (FTIR) analysis. These techniques are helpful to describe the size, shape and functional features of nanoparticles (23). Besides this, green synthesized AgNPs can be helpful in enhancing shelf life of banana in lower concentration without effecting its nutrition. 

 Page 3 and line 84-85

24. Reviewer 2 2- In the sample collection part: add the age of the used trees for more clarity.

 Young Eucalyptus leaves from around 40 years old eucalyptus tree were collected from GT Road near Pindi Bypass, Gujranwala produce silver nanoparticles. Page 3 and line 94-95

25. Reviewer 2 3- Add the manufacturer and the country for the spectrophotometer.

 UV-VIS Spectrometry

It is a reliable analytical technique which was used to evaluate the synthesis and functional stability. For this process dried nanoparticles were re-suspended in distilled water by recording spectra between 300-700 nm at 0.1 nm resolution through UV-VIS spectrophotometer (UV-1800 SHIMADZU, Shimazdu, Japan) (26).

Fourier transform infrared spectroscopy (FTIR)

 The FTIR technique was used to study the stability and synthesis of nano-scale silver particle by monitoring the 10 mg dry sample of AgNPs at resolution power of 4 at range of 500-4000 cm−1 infrared through Fourier-transform infrared spectrophotometer (NICOLET iS5, Thermo Scientific, USA) (27).

Energy dispersive x-ray analysis (EDX)

This analytical method was used to identify the desired elements. The spectra of peaks showed the true composition of sample by EDX with SEM (JSM-5910, INCA200 Oxford instruments, UK) (31).

 Page 4-5 and line 119, 124, 142

26. Reviewer 2 4- In the part of measuring starch content: what do you mean with hot ethanol? Ethanol evaporates at 60°C. Please more clarification is needed. To study the starch conversion, anthrone test was conducted. For this reason, the 0.5g banana pulp was homogenized with 5ml 80% ethanol in air tight beaker and incubated in water bath for 30 minutes at 80°C. Then it was centrifuged at 44000 rpm for 5 minutes. After that, 20 mL of distilled water and 6.5mL of perchloric acid was added in it. Then centrifuged at 4°C for 20 minutes and the supernatant was saved as extract. Page 7 and line 184-186

27. Reviewer 2 5- In the statistical analysis part: add the country of the used software and explain about the PCA The experiment was carried out in a completely randomized design (CRD) in which each treatment was directed in three replications. In order to study the effect of each treatment, the data was subjected to ANOVA test at 5% level of significance and these statistical tests were subjected by using the “Minitab 19” software (Originated at Pennsylvania State University, USA). Means were separated by the Duncan’s multiple range test. PCA (principal component analysis) was done to reduce multidimensional dataset.

 Page 8 and line 225-227

28. Reviewer 2 6- Since you know from beginning that the silver might inhibit the ethylene production in banana, why you did not measure ethylene from fruit? More explanation is needed. The parameter was in progress. Now added

Effect of AgNPs on ethylene production (ppm) in banana

Ethylene plays an important role in the ripening of banana. In addition to increase the process of fruit ripening, ethylene frequently causes over-ripening and even rotting, which shortens shelf life of fruits and vegetables. Maximum increase in ethylene production was recorded during ripening of uncoated banana sample as it was 36.5 ppm on day 0 which reached 77.5 ppm on day 14 with maximum decay and 1129.2 ppm on day 32. This could be due to high respiration rate and autocatalytic ethylene production that cause the physiological and metabolic changes by change in chloroplast structure that reduce the chlorophyll content and increase decay in uncoated bananas (85). Rather than control banana sample, treated banana depicted the less increase in ethylene rate from day 0 to day 32. All treated banana sample showed ethylene rate in rang of 35 to 38 ppm at day 0 while its production rate was 68.5 ppm with 0.01%, 70.3 ppm with 0.02%, 87.6 ppm with 0.03%, 89.4 ppm with 0.04% and 91.9 ppm with 0.05% AgNPs concentrations (Fig 11). Different concentrations of edible coating with chitosan nanoparticles suppressed the ethylene production and increase the shelf life of banana till 30 days of storage (38). Similarly, another study investigated the use of guar gum based AgNPs to accelerate the shelf life of mango for 28 days at 25oC by reducing the ethylene production (59). So, the AgNPs as fruit coating treatments act as a semipermeable membrane that reduce the respiration rate and ethylene production by altering internal atmosphere. It delay metabolic activity and potentially reduce the ripening process which ultimately lead to the increase in fruit storage life (59). 

 Page 19 and line 551-568

29. Reviewer 2 7- Add the significant effect between treatments to the text and as letters to the figure in all parts for more clarity All figures were changed into time-based bar graphs with treatments and statistical differences a, b and c has been added. In separate figure file as per format

30. Reviewer 2 8- In the part of “3.7.3. Effect of AgNPs on Banana Firmness (N)” correct the typo mistake for “understand” in line 9. The results are in accordance with the study that used the guar gum based AgNPs with carboxymethyl cellulose to enhance the shelf life of mango for 14 days at 25oC after the spoilage of untreated mango over the total storage period of 28 days Page 12 and line 356

31. Reviewer 2 9- It was unclear how the control fruit had the highest weight loss and the highest moisture content in the same time compared with other treatments! Please, more clarification is needed. Rechecked data and corrected

Effect of AgNPs on weight loss (%) of banana 

Weight loss is an important element to determine the quality of banana during prolonged storage period and to investigate the increase in the shelf life of banana. The results showed the weight loss percentage range of cavendish banana was 3.2-4.3% during storage time of 7 days that depends on the size of banana. During 14 days of storage, the weight loss percentage slightly increased in all banana samples at the range of 5.8- 6.6% except control that was 9.3%. Similarly, the weight loss percentage was 13.9% in control banana after 21 days, 21.5% at 28 days and 33% at 32 days of storage and they became non-edible. This could be linked to the reduced moisture retaining ability of un treated banana due to accelerated rate of transpiration and respiration from banana surface as well as deterioration in tissues of banana peel (62), while the weight loss percentage showed less increase as 10.8%, 12.8%, 12.8%, 11.6% and 16.8% with 0.01%, 0.02%, 0.03%, 0.04% and 0.05% AgNPs respectively at 32 days of storage (Fig 6B). The results are similar to the previous work where AgNPs synthesized from tea extract increased the weight loss percentage with increase in storage period in cherry tomatoes for 15 days (63). The increase in weight loss percentage with storage time was due to AgNPs coatings that act as semipermeable barriers which hinders the oxygen, carbon dioxide, water loss, respiration and other oxidation reactions and maintain the weight loss percentage in fruits by improving their post-harvest quality (6,58).

Page 12-13 and line 363-377

32. Reviewer 2 

10- In part of “7.7. Effect of AgNPs on pH of Banana” I suggest to delete the first sentence. First sentence has been deleted 

Effect of AgNPs on pH of banana

Banana pulp pH was influenced by different treatments of AgNPs. Normally, it decreases at the start of the ripening stage and continuously increase until it attains a fully ripen stage (69). Untreated banana showed pH values in range of 5.3-5.6 along with all banana samples treated with different concentrations of AgNPs at stage 0 to 7. It was noted that the control banana sample of Basrai started decay at pH 6.0 on day 21 and showed complete decay at pH 6.3 on day 28. The sharp increase in pH was due to change in metabolic process and less acidity that is directly proportional to respiration rate in fruit (70). In comparison to control, all treated banana samples maintain pH till 32 days of storage with slight change. Treated banana depicted the less change in pH as 5.6 with 0.01% and 5.5 with 0.02% among all treatments at 32 days of storage (Fig 8A). The results showed similarities with previous literature in which AgNPs maintain the pH value during the 30 days storage period of loquat at 4oC (70) and 10 days storage period of carrot at 10oC (71) while the sharp increase was seen in control samples and the fluctuations in results could be due to different climate conditions and type of fruit (70). Generally, pH indicated the amount of organic acids in pulp of banana, which decreased at ripened stage of untreated banana due to use of these organic acids for respiration. However, bananas treated with AgNPs maintained the pH throughout the storage period. The acidity at un-ripen stage affects the taste of banana due to partial presence of oxalic acid which de-carboxylate at ripening stage by oxalate oxidase. So, control banana samples showed more change in pH as compared to treated banana samples. Similarly, different varieties of banana showed changes in pH depending on their ripeness and experimental conditions (33,69,70).

 Page 14 and line 426

33. Reviewer 2 11- It was not clear to me how TA increased during shelf life and usually the TA decreased by increasing the fruit maturity and senescent. Please more clarification is needed. Rechecked data and corrected

Effect of AgNPs on titrable acidity (%) in banana 

Change in titrable acidity is linked to decay of banana. The change in titratable acidity of banana pulp was studied by applying various concentration of AgNPs beside untreated bananas. A gradual increase in titrable acidity was detected in all treatments over the storage period of 21 days, where decline was noted after 21 days till 32 days of storage. Titrable acidity ranged from 0.32-0.33% in treated banana at day 0, that gradually increased with the increase of pH during 7 days of storage except untreated banana samples. It was observed that the untreated bananas showed decrease in titrable acidity making banana non-edible in 28 days of storage with maximum change in titrable acidity that was 0.16% while banana treated with AgNPs maintained the post-harvest quality of banana till 32 days storage period. It could be linked to the rise in malic acid, citric acid, and oxalic acid with start of ripening. However, main cause of the decline in acidity of banana at maturity was conversion of acid into sugar content (72). Among all, lower change in titarable acidity was recorded with 0.01% ,0.02% and 0.03% AgNPs which was 0.33%, while gradual decrease in titrable acidity was recorded with 0.04% and 0.05% AgNPs that was 0.32% from day 0 to day 32 (Fig 8B). Similarly in mango fruits coated with guar gum based AgNPs recorded less change in titrable acid values while untreated fruits had significantly more change at the end of the storage period (59). In cherry tomatoes with Oolong tea-AgNps application showed stable acid contents at the end of 15 days storage as compared to chemically prepared AgNPs that showed only 3 days increase in shelf life (63). The fluctuation in results was due to the application of silver nanoparticle coatings that is responsible to maintain acid contents in banana during storage (70). Rapid increase in acid accumulation at immature stage raised the titrable acidity during ripening. Meanwhile the formation of sugar contents and physiological processes minimize the excessive increase of organic acids during maturity of banana (33,72,73).

 Page 14-15 and line 444-464

34. Reviewer 2 

12- add the “doi” to all references. “doi” has been added in all references In reference section

---

## [Decision Letter · Decision Letter 1]

30 Jan 2023

Green Synthesized Silver Nanoparticles from Eucalyptus Leaves can Enhance Shelf life of Banana without Penetrating in Pulp

PONE-D-22-24409R1

Dear Dr. Akhtar,

We’re pleased to inform you that your manuscript has been judged scientifically suitable for publication and will be formally accepted for publication once it meets all outstanding technical requirements.

Kind regards,

Mohammad Irfan, Ph.D.

Academic Editor

PLOS ONE

Additional Editor Comments (optional):

Reviewers' comments:

Reviewer's Responses to Questions

**Comments to the Author**

1. If the authors have adequately addressed your comments raised in a previous round of review and you feel that this manuscript is now acceptable for publication, you may indicate that here to bypass the “Comments to the Author” section, enter your conflict of interest statement in the “Confidential to Editor” section, and submit your "Accept" recommendation.

Reviewer #1: All comments have been addressed

Reviewer #2: All comments have been addressed

2. Is the manuscript technically sound, and do the data support the conclusions?

Reviewer #1: Yes

Reviewer #2: Yes

3. Has the statistical analysis been performed appropriately and rigorously? 

Reviewer #1: Yes

Reviewer #2: Yes

4. Have the authors made all data underlying the findings in their manuscript fully available?

Reviewer #1: Yes

Reviewer #2: Yes

5. Is the manuscript presented in an intelligible fashion and written in standard English?

Reviewer #1: Yes

Reviewer #2: Yes

6. Review Comments to the Author

Reviewer #1: The revised manuscript has been evaluated . I would like to appreciate the authors for addressing the major concerns regarding this manuscript. Still there are some minor changes which have been added in the attached PDF file. The manuscript can be accepted for publication in PLos One.

Reviewer #2: No comment.

Thank you for following the comments and suggestions from the previous revision.

All best

7. PLOS authors have the option to publish the peer review history of their article (what does this mean?). If published, this will include your full peer review and any attached files.

Reviewer #1: **Yes: **Bhavya Bhargava

Reviewer #2: No

---

## [Editor Report · Acceptance letter]

27 Feb 2023

PONE-D-22-24409R1 

Green Synthesized Silver Nanoparticles from Eucalyptus Leaves can Enhance Shelf life of Banana Without Penetrating in Pulp 

Dear Dr. Akhtar:

I'm pleased to inform you that your manuscript has been deemed suitable for publication in PLOS ONE. Congratulations! Your manuscript is now with our production department. 

Kind regards, 

on behalf of

Dr. Mohammad Irfan 

Academic Editor

PLOS ONE